# Hokoff: Real Game Dataset from Honor of Kings and its Offline Reinforcement Learning Benchmarks

**Yun Qu**[*†], **Boyuan Wang**[*†], **Jianzhun Shao**[*†], **Yuhang Jiang**[†], **Chen Chen**[†], **Zhenbin Ye**[♮], **Lin Liu**[♮], **Junfeng Yang**[♮], **Lin Lai**[♮], **Hongyang Qin**[§], **Minwen Deng**[§], **Juchao Zhuo**[§], **Deheng Ye**[§], **Qiang Fu**[§], **Wei Yang**[§], **Guang Yang**[♮], **Lanxiao Huang**[♮], **Xiangyang Ji**[†]

[†]Tsinghua University, [♮]Tencent Timi Studio, [§]Tencent AI Lab
{qy22,wangby22,sjz18,jiangyh19}@mails.tsinghua.edu.cn,chenchen.peach@gmail.com,
{zhenbinye,lincliu,fengjunyang,linlai,hongyangqin,danierdeng,jojozhuo,dericye,leonfu,
willyang,mikoyang,jackiehuang}@tencent.com,xyji@tsinghua.edu.cn

## Abstract

The advancement of Offline Reinforcement Learning (RL) and Offline Multi-Agent Reinforcement Learning (MARL) critically depends on the availability of high-quality, pre-collected offline datasets that represent real-world complexities and practical applications. However, existing datasets often fall short in their simplicity and lack of realism. To address this gap, we propose `Hokoff`, a comprehensive set of pre-collected datasets that covers both offline RL and offline MARL, accompanied by a robust framework, to facilitate further research. This data is derived from Honor of Kings, a recognized Multiplayer Online Battle Arena (MOBA) game known for its intricate nature, closely resembling real-life situations. Utilizing this framework, we benchmark a variety of offline RL and offline MARL algorithms. We also introduce a novel baseline algorithm tailored for the inherent hierarchical action space of the game. We reveal the incompetency of current offline RL approaches in handling task complexity, generalization and multi-task learning.

## 1 Introduction

Online Reinforcement Learning (Online RL) relies on the interaction between the training policy and the environment for data collection and policy optimization [33, 18, 7]. However, this paradigm makes online RL unsuitable for certain real-world scenarios, such as robotics and autonomous driving [23, 18], as deploying untested policies to the environment can be costly and dangerous [19]. In contrast, Offline Reinforcement Learning (Offline RL) can learn satisfactory policies using a fixed dataset without the need for further interaction with the environment [23, 18, 7, 19]. This characteristic alleviates the aforementioned issue, making offline RL potentially more suitable for certain real-world scenarios compared to online RL [23].

The research on offline RL has attracted significant attention in recent years and has made substantial progress in both theoretical analysis and practical performance. The core challenge of offline RL is the value overestimation issue induced by distributional shift [9, 21]. Existing studies mitigate this problem by constraining the learning policy to closely resemble the behavior policy induced by the dataset [39, 18, 7], or adopting conservative value iteration [19, 16]. The success of offline RL can be largely attributed to the availability of widely-adopted open-access datasets, such as D4RL [6] and RL Unplugged [12]. These datasets offer standardized and diverse pre-collected data for the development of new algorithms, while also offering proper evaluation protocols that facilitate fair comparisons between different algorithms. However, despite their benefits, tasks contained in these datasets (such

---

[*] Authors contributed equally

as Atari 2600 [3] or Mujoco [35]) are often overly simplistic or purely academic, failing to simulate the complexity of real-world scenarios and lacking practical applications. Thus, there is a significant gap between offline RL research and its practical application in real-world settings. This disparity hinders the usefulness of offline RL in addressing real-world problems, and thus, it is indispensable to create datasets that reflect a realistic level of complexity and practicality in real-world applications.

Offline Multi-Agent Reinforcement Learning (Offline MARL) [28, 43] is gaining increasing attention due to its close relationship to real scenarios, such as games [31, 38], sensor networks [49] and autonomous vehicle control [42]. However, the lack of standardized datasets restricts the development 40 of offline MARL. Existing works only rely on self-made datasets, which hampers fairness and reproducibility. Moreover, the settings they focus on are typically limited to toy examples (e.g., Multi-agent Particle Environment [26]) or simplified versions of classic games (e.g., StarCraft Multi-Agent Challenge [31]), facing the same impractical issues encountered in offline RL. Thus, there is an urgent need for open-access datasets to further the progress of offline MARL.

The connections between offline RL and offline MARL are closely linked due to similar challenges pertaining to offline learning. While, offline MARL also introduces distinct algorithm development requirements, as it involves unique characteristics like multiple agents and intra-team cooperation. The current offline MARL algorithms [28, 43] are mainly adaptations of offline RL algorithms, necessitating the availability of standardized offline datasets that cater to both single-agent and multi-agent settings. Thus, to enhance versatility and practicality, it is crucial to propose datasets that encompass both single-agent settings and multi-agent settings.

In this paper, we present `Hokoff`, a suite of pre-collected datasets for both offline RL and offline MARL, along with a comprehensive framework for conducting corresponding research. Our paper makes several novel contributions, which are shown below:

• The tasks we adopt are based on one of the world's most popular Multiplayer Online Battle Arena (MOBA) games, Honor of Kings (HoK), which has over 100 million daily active players[38], ensuring the practicality of our datasets. The complexity of this environment dramatically surpasses those of its counterparts, demonstrating the potential for simulating real-world scenarios.

• We present an open-source, easy-to-use framework [1] under Apache License V2.0. This framework includes comprehensive processes for offline RL (sampling, training, and evaluation), and some useful tools. Based on the framework, We release a rich and diverse set of datasets [2] which are generated using a series of pre-trained models featuring distinct design factors. These datasets cater not only to offline RL but also offline MARL.

• Building on the framework, we reproduce various offline RL and offline MARL algorithms and propose a novel baseline algorithm tailored for the inherent hierarchical structured action space of Honor of Kings. We fully validate and compare these baselines on our datasets. The results indicate that current offline RL and offline MARL approaches are unable to effectively address complex tasks with discrete action space. Additionally, these methods exhibit shortcomings in terms of their generalization capabilities and their ability to facilitate multi-task learning.

## 2 Related Works

### 2.1 Offline RL and Offline MARL

Offline RL gains significant attention in recent years, primarily due to the inherent difficulties of directly applying online RL algorithms to offline environments. The main hurdles encountered is the issue of erroneous value overestimation, which arises from the distributional shift between the dataset and the learning policy [9]. Theoretical studies have demonstrated that the overestimation issue can be alleviated by pessimism, which results in satisfactory performance even with imperfect data coverage [4, 5, 14, 20, 25, 30, 41, 48]. In practice, certain studies [1, 2, 40, 1, 2, 40] employ uncertainty-based methods to estimate Q-values pessimistically or to perform learning on pessimistic dynamic models by estimating the epistemic uncertainty of Q-values or dynamics. Some studies [18, 9, 39, 16, 19, 7] adopt behavior regularization-based approaches by imposing constraints on the

---

[1] `https://github.com/tencent-ailab/hokoff`
[2] `https://sites.google.com/view/hok-offline`

learned policy to align closely with the behavior policy, either explicitly or implicitly, which offers better computational efficiency and memory consumption compared to uncertainty-based methods.

Offline MARL, combining offline RL and MARL, has emerged in recent years to address safety and training efficiency concerns in practical multi-agent scenarios. Most studies in this domain adopt a multi-agent paradigm, such as independent learning [34] or centralized training with decentralized execution (CTDE) [26]. These investigations also incorporat offline methods, similar to those employed in single-agent settings, to mitigate distributional shift. Moreover, innovative treatments are introduced for cooperation, such as zeroth-order optimization in OMAR [28] or decomposing the joint-policy in MAICQ [43]. In addition, Jiang & Lu [13] specifically focuses on decentralized learning using BCQ [9], while Tseng et al. [36] regards offline MARL as a sequence modeling problem, utilizing supervised learning and knowledge distillation to tackle the challenges it presents.

## 2.2 Offline Datasets

The availability of large-scale pre-collected datasets has greatly facilitated the progress of deep supervised learning [11]. Offline RL, which is regarded as a bridge between RL and supervised learning, also requires learning policies from pre-collected datasets [6]. Therefore, high-quality pre-collected offline datasets play a significant role in the development of offline RL. To meet this demand, some datasets have been published and widely adopted. D4RL [6] is designed to address key challenges often faced in practical applications where datasets may have limited and biased distributions, incomplete observations, and suboptimal data. To tackle these issues, D4RL offers a range of datasets that enjoy these characteristics. Similarly, RL Unplugged [12] introduces a benchmark to evaluate and compare offline RL methods with various settings, such as partially or fully observable and continuous or discrete actions. These offline datasets play a significant role in offline RL research, and many previous works train and evaluate their methods based on these datasets [16, 2, 15, 28, 43].

However, both D4RL and RL Unplugged primarily focus on relatively simple tasks and lack high-dimensional, practical and multi-agent tasks that closely resemble real-world scenarios. StarCraft II Unplugged [27] introduces a benchmark for StarCraft II, a complex simulated environment with several practical properties. However, they only utilize a dataset derived from human replays, which lacks diversity in design for offline RL, and they did not evaluate existing offline RL methods. To address this research gap, we propose `Hokoff`, a benchmark based on HoK, which aims to provide diverse offline datasets for high-dimensional, practical tasks, and present a comprehensive evaluation of previous offline RL and offline MARL methods with a general, easy-to-use framework.

## 3 Background

Honor of Kings (HoK) is one of the most popular MOBA games worldwide, boasting over 100 million daily active players [38]. The game involves two teams, each consisting of several players who have the option to select from a wide range of heroes with diverse roles and abilities. In the game, heroes are expected to eliminate enemy units, such as heroes, creeps, and turrets, to gain gold and experience. The primary objective is to destroy the enemies' turrets and crystal while defending their own. To succeed in MOBA games, players must learn how to choose the appropriate hero combination, master complex information processing and action control, plan for long-term decision-making, cooperate with allies, and balance multiple interests. The complex rules and properties of HoK make it be more in line with the complex decision-making behavior of human society. Thus, HoK has attracted numerous researchers interest [45, 44, 38, 37, 10].

The underlying system dynamic of HoK can be characterized by a Partially Observable Markov Decision Process (POMDP [33]), denoted by $\mathcal{M} = (\mathcal{S}, \mathcal{O}, \mathcal{A}, \mathcal{P}, r, \gamma, d)$. Due to the fog of war and private features, each agent has access to only local observations $\boldsymbol{o}$ rather than the global state $s$. Specifically, the agents are limited to perceiving information about game units within their field of view, as well as certain global features. Due to the intricate nature of control, the action space $\mathcal{A}$ is organized in a hierarchically structured manner, rather than being flattened, which avoids the representation of millions of discretized actions. Randomness is added into the transition distribution $\mathcal{P}$ in the form of critical hit rate. The reward $r$ is decomposed into *multi-head* form and each hero's reward is a weighted sum of different reward items and is designed to be *zero-sum*. Details of observation space, action space and reward are presented in Appendix D.

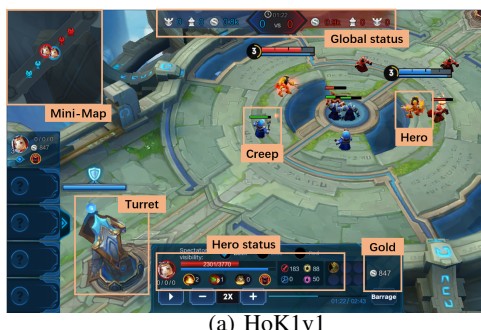 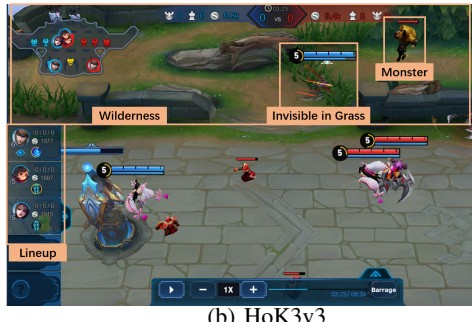

| (a) HoK1v1 | (b) HoK3v3 |

Figure 1: (a) The Game replay user interface (UI) in HoK1v1. (b) The UI in HoK3v3. Important information and units of the game are highlighted using orange boxes.

## 4 Hokoff

This study is based on the HoK gaming environment, which encompasses both 1v1 and 3v3 maps. Our research proposes a comprehensive offline RL framework applicable to this gaming environment and utilizes it to generate diverse datasets. This section provides an introduction to the framework, game modes, datasets, and evaluation protocol employed in this study.

### 4.1 Framework

To enhance the usability of our `Hokoff`, we propose a reliable and comprehensive Offline RL framework that consists of three modules: sampling, training, and evaluation. This framework streamlines the process of sampling new datasets, developing and training baselines, and evaluating their performance. The sampling module provides a simple and unified program for sampling diverse datasets using any pre-trained checkpoints. There are several reasons why our framework excels in sampling. Firstly, diverse datasets at different levels of expertise can be sampled by leveraging Multi-Level Models as described in Sec. 4.1.1. Secondly, our framework employs parallel sampling techniques, ensuring efficient sampling of large and diverse datasets. Based on the training module, we have implemented various offline RL and offline MARL algorithms as baselines. Additionally, we consolidate crucial components and provide user-friendly APIs, facilitating researchers to effortlessly develop novel algorithms or innovative network architectures. The evaluation module enables the assessment of trained models from different algorithms, ensuring fair comparisons. Fig. 2 demonstrates the architecture of our framework and Appendix E provides an example of the APIs.

#### 4.1.1 Multi-Level Models

To ensure a valid and unbiased comparison of the performance of distinct algorithms, it is crucial to establish appropriate evaluation protocols [6, 12]. One such effective evaluation protocol is the normalized score [6]. However, HoK is a zero-sum adjustable rewards MOBA game. The episode return in the game is heavily influenced by the opponents and game settings, and the objective is to win, which renders the use of return as a performance metric biased. Therefore, normalized score may not fully capture our requirements. Furthermore, similar to our situation, the evaluation protocol for SMAC [31], a competitive game, is based on win rate against a pre-programmed AI. Nonetheless, it is exceedingly challenging to create a built-in AI with human-like performance due to the complexity of MOBA games.

Inspired by prior works of HoK [38], we present Multi-Level Models for sampling and evaluating which contains multiple checkpoints with different level. Specifically, we have extracted several checkpoints from pre-trained dual-clip PPO [45, 44] models with varying levels determined by the outcome of the battle separately for HoK1v1 and HoK3v3. We adopt the *win rate* against different checkpoints as our evaluation protocols to assess the ability of models. Additionally, these models, with varying levels, can be utilized on both sides to sample diverse battle data. The capabilities of these models surpass those of rule-based AI and match the levels of different human players, thus making these evaluation protocols more suitable for comparing algorithmic performance with human-level performance and facilitating diverse and effortless sampling. The details of these Multi-Level Models are provided in the Appendix F.

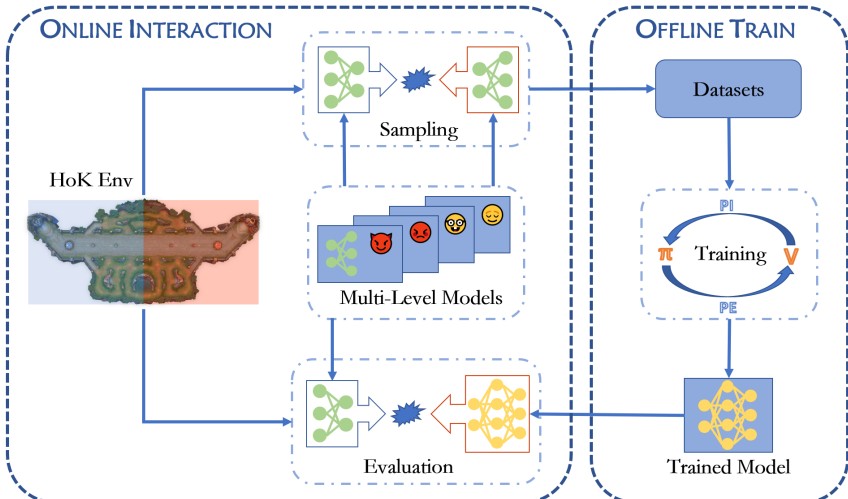

Figure 2: The architecture of the framework. The sampling and evaluation modules should interact with the environment. Multi-Level Models are the foundation baseline models of these two modules, serving as opponents in the evaluation module and being on both sides in the sampling module, as described in Sec 4.1.1. The training module is responsible for training offline RL algorithms using fixed datasets and producing trained models for evaluation.

## 4.2 Game Modes

We have incorporated two game modes from HoK into our study, namely HoK1v1 [38] and HoK3v3. The environment code of HoK3v3 is integrated into the open-source HoK1v1 code[3], following Apache License V2.0. These game modes differ in the number of agents involved and the underlying map used. Detailed information on each game mode is presented below.

### 4.2.1 Honor of Kings Arena

Honor of Kings Arena (HoK Arena or HoK1v1) is a 1v1 game mode where each player attempts to beat the other and destroy its opponent's crystal. Specifically, each player chooses a hero before the game starts and controls it during the whole game. There are a total of 20 heroes available for players to select, each possessing distinct skills that exert diverse effects on the game environment. The observation space is a continuous space consisting of 725 dimensions that contain partial observable information about the hero, opponent, and other game units. The action space is hierarchically structured and discretized, covering all possible actions of the hero in a hierarchical triplet form: (1) which action button to take; (2) who to target; and (3) how to act. Furthermore, the reward is a weighted sum of five categories: farming, kill-death-assist (KDA), damage, pushing, and win-lose. For a full description of this game mode, please refer to the Appendix D.1.

### 4.2.2 Honor of Kings 3v3 Arena

To further cater to the demand for Offline MARL, we adopt Honor of Kings 3v3 Arena (HoK3v3) as our experimental platform. HoK3v3 is a MOBA game, where each team comprises three heroes who collaborate to defeat their opponents. The basic rules and win conditions of HoK3v3 are similar to HoK1v1. However, the HoK3v3 map contains additional turrets on the middle road and features a new area called the "wilderness", inhabited by diverse monsters. Besides, collaboration is essential in HoK3v3, where players must select different heroes and fulfill distinct roles to work together more efficiently. For instance, one hero might focus on slaying monsters in the wilderness to earn gold and experience, while the other heroes engage in offensive tactics against the enemy heroes and game units. The design philosophies for observation space, action space, and reward are comparable to those used in HoK1v1. However, the level of complexity in HoK3v3 is significantly elevated. We provide a detailed description of the game mode in the Appendix D.2 for reference.

---

[3]`https://github.com/tencent-ailab/hok_env`

### 4.2.3 Subtasks

Both HoK1v1 and HoK3v3 are full MOBA games, featuring multi-camp competitions, which inherently pose challenges and limitations. Consequently, training on these game modes demands extensive training time and computational resources. However, HoK game comprises various sub-objectives, allowing us to decompose the overall game into manageable subtasks. These subtasks can represent diverse scenarios and are suitable for evaluating various algorithms. In this study, we propose two specific noncompetitive subtasks as outlined below. It is worth noting that researchers can readily expand upon our framework to develop additional subtasks.

***Destroy Turret:*** One of the key sub-objectives in HoK is to destroy the enemy's turrets as quickly as possible, to gain access to the enemy crystal. To train this specific skill, we have devised a subtask called *Destroy Turret*, which is based on HoK1v1. In this subtask, the focus is solely on destroying the enemy's turret and crystal as quickly as possible, and the enemy hero is removed.

***Gain Gold:*** Gold is a critical resource in HoK that can be used to purchase equipment, which enhances the abilities of the heroes. Inspired by resource collection tasks from previous studies [22], we have designed a subtask called *Gain Gold*, which is based on HoK3v3, where the new objective is to collect golds in restricted time steps, and the enemy heroes are removed. As a multi-agent setting, it focuses on the cooperation or intra-team competition while avoiding inter-team competition.

## 4.3 Datasets

To enhance the practical implications of our datasets, we have incorporated design factors that align with the real-world applications of both HoK and other relevant scenarios.

***Multi-Difficulty***

Intuitively, the level of difficulty in the environment significantly impacts the performance of algorithms. However, previous researches only utilized one set of datasets with a uniform level of difficulty in the environment, which is not appropriate for HoK, where the difficulty of the task can be substantially affected by the level of opponents. Therefore, to examine the effects of varying levels of difficulty in the environment, we propose several multi-difficulty datasets with different difficulty levels. Specifically, we develop two sets of datasets: *norm* and *hard*, which are categorized based on the opponent's level. Within each level, we propose four datasets according to diverse win rates against the opponent: *poor, medium, expert* and *mixed*. To elaborate, the *poor/medium/expert* dataset is generated by recording the battle trajectories of a relative *lower/equal/higher* level model compared to the opponent, and the *mixed* dataset is an equal mixture of the three datasets mentioned above.

***Multi-Task***

As a MOBA game, HoK features a diverse cast of heroes with distinct roles and skillsets. While the overall objective remains consistent throughout matches, the selection of heroes can significantly alter the nature of the task at hand. Consequently, HoK presents multi-task challenge which requires a single model to handle multiple tasks [47, 24, 46]. However, none of the current works provide uniform datasets for multi-task offline RL. To address this research gap, we propose a series of multi-task datasets based on the multi-task nature of HoK and evaluate the multi-task learning ability of current offline RL and offline MARL algorithms. Specifically, we define a hero pool with several heroes and randomly select heroes from it to sample data. Depending on whether the selected heroes are on the controlled side or the opponent side, we sample either the *multi_hero* or the *multi_oppo* dataset. In cases where both sides choose random heroes, we sample the *multi_hero_oppo* dataset.

Furthermore, as mentioned in the previous section, different levels of opponents naturally form multiple tasks with varying environmental difficulties. Thus, we propose several level-based multi-task datasets by sampling data with randomly selected opponent levels. According to different difficulty levels, we have proposed two datasets, named *norm_multi_level* and *hard_multi_level*.

***Generalization***

The unique gameplay mechanics of HoK, characterized by a diverse cast of heroes with distinct roles and skillsets, lend themselves well to multi-task and serve as an ideal testbed for evaluating the generality of models across a range of tasks. Building on the previous work [38] and taking into account the realities of human combat in HoK, we have identified three key challenges for generalization: hero generalization, opponent generalization, and level generalization.

We have developed six experiments: "norm_general" and "hard_general" for level generalization, "norm_hero_general" and "hard_hero_general" for hero generalization, and "norm_oppo_general" and "hard_oppo_general" for opponent generalization, for HoK1v1 and HoK3v3, respectively. Among them, the first two experiments, "norm_general" and "hard_general," have their corresponding datasets, and we train models on these datasets. The latter four experiments do not require extra datasets because we directly use the existing models that have already been trained using other datasets. For more details on the design of generalization, please refer to the Appendix C.

*Heterogeneous Teammate*

Heterogeneous teammate is a crucial research direction in MARL [32, 17]. In the practical scenarios of HoK, the capacity of each player is generally different, making it naturally suitable for investigating the challenges associated with heterogeneous teammate. In order to mimic real-world scenarios and facilitate research on heterogeneous teammate challenges, we design two datasets in HoK3v3: *norm_stupid_partner* and *norm_expert_partner*. These datasets were collected in a standard manner, with the exception that one random hero in each team is controlled by a model with a relatively low/high level of expertise, while the remaining heroes are controlled by the regular model.

*Sub-Task*

As introduced in Sec 4.2.3, we designed several practical and meaningful sub-tasks to provide diverse scenarios based on HoK. Based on these sub-tasks, we proposed diverse datasets to support Offline RL research similar to the design of previous studies [6, 12].

Table 1: Details of datasets in HoK1v1 game mode

| Factors | Datasets/Experiments | Capacity | Heroes | Oppo_heroes | Win_rate | Levels |
|---|---|---|---|---|---|---|
| **Multi-Difficulty** | **norm_poor** | 1000 | default | default | 12% | 1 |
| | **norm_medium** | 1000 | default | default | 50% | 1 |
| | **norm_expert** | 1000 | default | default | 88% | 1 |
| | **norm_mixed** | 1000 | default | default | 50% | 1 |
| | **hard_poor** | 1000 | default | default | 6% | 5 |
| | **hard_medium** | 1000 | default | default | 50% | 5 |
| | **hard_expert** | 1000 | default | default | 84% | 5 |
| | **hard_mixed** | 1000 | default | default | 45% | 5 |
| **Generalization** | **hard_general** | 1000 | default | default | 90% | 5 |
| | **norm_general** | 1000 | default | default | 46% | 1 |
| | **norm_hero_general** | - | multi_hero | default | - | 1 |
| | **hard_hero_general** | - | multi_hero | default | - | 5 |
| | **norm_oppo_general** | - | default | multi_hero | - | 1 |
| | **hard_oppo_general** | - | default | multi_hero | - | 5 |
| **Multi-Task** | **norm_multi_level** | 1000 | default | default | 50% | 1 |
| | **hard_multi_level** | 1000 | default | default | 50% | 5 |
| | **norm_multi_hero** | 1000 | multi_hero | default | 23% | 1 |
| | **norm_multi_oppo** | 1000 | default | multi_hero | 77% | 1 |
| | **norm_multi_hero_oppo** | 1000 | multi_hero | multi_hero | 50% | 1 |

### 4.3.1 Datasets Details

Table 1, Table 2 and Table 3presents the details of our proposed datasets. All the datasets are sampled using checkpoints with different levels as introduced in Sec. 4.1.1. Typically, each dataset consists of 1000 trajectories, except for the sub-task datasets, which contain 100 trajectories. The default heroes chosen for both camps are *luban* with Summoner Spells set to *frenzy* in HoK1v1 and *{{zhaoyun}, {diaochan}, {liyuanfang}}* with Summoner Spells assigned as *{{smite}, {purify}, {purify}}* based on their respective roles in HoK3v3. However, in specific scenarios such as *Generalization* or *Multi-Task* settings, we employ a random selection of heroes from a predefined set, *multi_hero*. For the HoK1v1 mode, the set comprises five heroes, *{luban, direnjie, houyi, makeboluo, gongsunli}*. In HoK3v3, the set consists six heroes, with two heroes assigned to each role, namely *{{zhaoyun, zhongwuyan}, {diaochan, zhugeliang}, {liyuanfang, sunshangxiang}}*. The win rate of the behavior policy is recorded in the column labeled *Win_rate* for reference. The column labeled *Levels* denotes the levels of opponents used for evaluation. More details of the datasets are presented in Appendix C.

Table 2: Details of datasets in HoK3v3 game mode

| Factors | Datasets/Experiments | Capacity | Heroes | Oppo_heroes | Win_rate | Levels |
|---|---|---|---|---|---|---|
| **Multi-Difficulty** | **norm_poor** | 1000 | default | default | 16% | 1 |
| | **norm_medium** | 1000 | default | default | 50% | 1 |
| | **norm_expert** | 1000 | default | default | 82% | 1 |
| | **norm_mixed** | 1000 | default | default | 49% | 1 |
| | **hard_poor** | 1000 | default | default | 18% | 7 |
| | **hard_medium** | 1000 | default | default | 50% | 7 |
| | **hard_expert** | 1000 | default | default | 83% | 7 |
| | **hard_mixed** | 1000 | default | default | 51% | 7 |
| **Generalization** | **hard_general** | 1000 | default | default | 94% | 8 |
| | **norm_general** | 1000 | default | default | 57% | 5 |
| | **norm_hero_general** | - | multi_hero | default | - | 1 |
| | **hard_hero_general** | - | multi_hero | default | - | 7 |
| | **norm_oppo_general** | - | default | multi_hero | - | 1 |
| | **hard_oppo_general** | - | default | multi_hero | - | 7 |
| **Multi-Task** | **norm_multi_level** | 1000 | default | default | 50% | 1 |
| | **hard_multi_level** | 1000 | default | default | 50% | 7 |
| | **norm_multi_hero** | 1000 | multi_hero | default | 74% | 1 |
| | **norm_multi_oppo** | 1000 | default | multi_hero | 26% | 1 |
| | **norm_multi_hero_oppo** | 1000 | multi_hero | multi_hero | 50% | 1 |
| **Heterogeneous** | **norm_stupid_partner** | 1000 | default | default | 50% | 1 |
| | **norm_expert_partner** | 1000 | default | default | 50% | 1 |
| | **norm_mixed_partner** | 1000 | default | default | 50% | 1 |

Table 3: Details of datasets in *Sub-Tasks*

| Sub-Task | Datasets/Experiments | Capacity | Heroes | Oppo_heroes | Average Score | Levels |
|---|---|---|---|---|---|---|
| **Destroy Turret** | **destroy_turret_medium** | 100 | default | no | 0.55 | medium |
| | **destroy_turret_expert** | 100 | default | no | 1.00 | expert |
| | **destroy_turret_mixed** | 100 | default | no | 0.73 | - |
| **Gain Gold** | **gain_gold_medium** | 100 | default | no | 0.13 | medium |
| | **gain_gold_expert** | 100 | default | no | 1.04 | expert |
| | **gain_gold_mixed** | 100 | default | no | 0.58 | - |

# 5 Benchmarking

Based on our framework, we reproduce various Offline RL and Offline MARL algorithms. Besides, we fully validate and compare these baselines on our datasets. The results are presented in the form of test winning rate. Each algorithm is run for three random seeds, and we report the mean performance with standard deviation. The performance of behaviour policies is presented in Appendix C. Details of the implementations and experimental results can be referenced in Appendix G.

## 5.1 Baselines

### 5.1.1 HoK1v1

The Offline RL baseline algorithms we implement are briefly introduced below: **BC**: Behavior cloning. **TD3+BC** [7]: One of the state-of-the-art single agent offline algorithm, simply adding the BC term to TD3 [8]. **CQL** [19]: Conservative Q-Learning conducts conservative value iteration by adding a regularizer to the critic loss. **IQL** [16]: Implicit Q-Learning leverages upper expectile value function to learn Q-function and extracts policy via advantage-weighted behavioral cloning.

The structured action space in HoK is similar to the joint action space in multi-agent settings, which inspires us to resort to the design in MARL methods. We propose a novel baseline algorithm, named **QMIX+CQL**. Specifically, we import QMIX algorithm from the MARL literature [29] to tackle the structured action space by regarding each head of the action space as a single agent and incorporate CQL regularizer term into local Q-funtion in QMIX for offline learning.

### 5.1.2 HoK3v3

The Offline MARL baseline algorithms are briefly introduced below: **IND+BC**: Behavior cloning with independent learning paradigm. **IND+CQL**: Adopts an independent learning paradigm for multi-agent settings, using conservative Q-learning [19]. **COMM+CQL**: Incorporate inter-agent communication based on IND+CQL. **IND+ICQ** [43]: Implicit Constraint Q-learning with independent learning paradigm, which only uses insample data for value estimation to alleviate the extrapolation error. **MAICQ** [43]: Multi-agent version of implicit constraint Q-learning by decomposed multi-agent joint-policy under implicit constraint with CTDE paradigm. **OMAR** [28]: Using zeroth-order optimization for better coordination among agents' policies, based on independent CQL.

Table 4: Averaged test winning rate or normalized score (*Sub-Task*) of baselines in HoK1v1 game mode.

| Factors | Datasets | BC | CQL | QMIX+CQL | IQL | TD3+BC |
|---|---|---|---|---|---|---|
| **Multi-Difficulty** | norm_poor | **0.08±0.02** | 0.06±0.01 | **0.08±0.02** | 0.07±0.01 | 0.0±0.0 |
| | norm_medium | **0.33±0.01** | 0.32±0.01 | 0.31±0.03 | 0.32±0.01 | 0.01±0.01 |
| | norm_expert | 0.64±0.01 | 0.58±0.03 | **0.67±0.01** | 0.62±0.02 | 0.03±0.01 |
| | norm_mixed | 0.17±0.01 | 0.23±0.04 | 0.20±0.01 | **0.25±0.01** | 0.01±0.01 |
| | hard_poor | 0.01±0.01 | 0.01±0.01 | 0.01±0.01 | 0.01±0.00 | 0.00±0.00 |
| | hard_medium | 0.13±0.01 | 0.11±0.01 | **0.20±0.01** | 0.12±0.02 | 0.00±0.00 |
| | hard_expert | 0.33±0.01 | 0.30±0.01 | **0.44±0.05** | 0.34±0.04 | 0.00±0.00 |
| | hard_mixed | 0.05±0.3 | 0.02±0.01 | **0.08±0.01** | 0.06±0.01 | 0.01±0.01 |
| **Generalization** | norm_general | 0.19±0.01 | 0.20±0.04 | **0.32±0.03** | 0.18±0.01 | 0.02±0.02 |
| | hard_general | 0.04±0.01 | 0.03±0.01 | **0.08±0.02** | 0.02±0.01 | 0.00±0.00 |
| | norm_hero_general | 0.06±0.01 | 0.06±0.01 | **0.08±0.01** | 0.07±0.01 | 0.00±0.00 |
| | hard_hero_general | 0.03±0.01 | 0.03±0.01 | 0.04±0.01 | **0.06±0.01** | 0.00±0.00 |
| | norm_oppo_general | **0.58±0.03** | 0.52±0.04 | 0.42±0.22 | 0.51±0.07 | 0.12±0.01 |
| | hard_oppo_general | 0.15±0.02 | 0.12±0.03 | **0.23±0.04** | 0.14±0.03 | 0.01±0.01 |
| **Multi-Task** | norm_multi_level | 0.32±0.03 | 0.25±0.03 | **0.41±0.02** | 0.30±0.02 | 0.02±0.01 |
| | hard_multi_level | 0.08±0.02 | 0.06±0.01 | **0.16±0.03** | 0.08±0.02 | 0.00±0.00 |
| | norm_multi_hero | 0.08±0.01 | 0.07±0.02 | **0.11±0.01** | 0.06±0.01 | 0.00±0.00 |
| | norm_multi_oppo | 0.59±0.02 | 0.55±0.03 | **0.65±0.02** | 0.60±0.05 | 0.10±0.02 |
| | norm_multi_hero_oppo | 0.26±0.01 | 0.21±0.02 | **0.32±0.03** | 0.28±0.05 | 0.03±0.01 |
| **Sub-Task** | destroy_turret_medium | 0.61±0.06 | 0.63±0.01 | 0.61±0.03 | 0.60±0.02 | **0.67±0.03** |
| | destroy_turret_expert | 0.94±0.02 | 0.94±0.02 | 0.92±0.05 | **0.95±0.01** | 0.57±0.13 |
| | destroy_turret_mixed | 0.88±0.04 | 0.87±0.03 | **0.89±0.02** | **0.89±0.04** | 0.82±0.03 |

## 5.2 Benchmark Results

We have validated the offline RL and offline MARL baselines on our datasets and aggregated the results in Table 4 and Table 5.

• **Baselines Comparison:** As indicated in Table 4, QMIX+CQL exhibits superior performance in comparison to other approaches, implying that the integration of MARL methods may be a suitable choice for environments with a structured action space. Moreover, in HoK3v3, IND+ICQ exhibits the highest performance across most datasets, except for the *Heterogeneous* datasets. Conversely, algorithms based on TD3, namely **TD3+BC** and **OMAR**, yield poor results.

• **Multi-Difficulty:** The baseline performance exhibits a significant decrease on the *hard-level* datasets compared with *norm-level* datasets, highlighting the limitations of current offline methods in addressing challenging tasks with discrete action space.

• **Generalization:** The disparities between training and evaluation in *Generalization* settings impede the achievement of desirable performance, indicating the inadequacy of current methods' generalization ability.

• **Multi-Task:** Training models on *Multi-Task* datasets results in a substantial performance enhancement compared to generalization settings. However, none of these models have been able to exceed the performance achieved by the behavior policy, underscoring the need for further research into the direct application of offline methods to multiple tasks.

Table 5: Averaged test winning rate or normalized score (*Sub-Task*) of baselines in HoK3v3 game mode.

| Factors | Datasets | IND+BC | COMM+CQL | IND+CQL | IND+ICQ | MAICQ | OMAR |
|---|---|---|---|---|---|---|---|
| **Multi-Difficulty** | norm_poor | 0.1±0.01 | 0.09±0.02 | 0.03±0.01 | **0.12±0.02** | **0.12±0.04** | 0.02±0.01 |
| | norm_medium | **0.48±0.01** | 0.47±0.04 | 0.4±0.03 | 0.45±0.01 | 0.38±0.16 | 0.23±0.03 |
| | norm_expert | 0.52±0.03 | 0.76±0.13 | **0.84±0.06** | 0.65±0.12 | 0.61±0.09 | 0.39±0.16 |
| | norm_mixed | 0.35±0.25 | **0.48±0.12** | 0.46±0.12 | 0.44±0.19 | 0.24±0.16 | 0.17±0.2 |
| | hard_poor | 0.16±0.03 | 0.11±0.04 | 0.12±0.03 | **0.17±0.02** | 0.12±0.03 | 0.08±0.04 |
| | hard_medium | 0.38±0.05 | 0.35±0.03 | 0.31±0.02 | **0.4±0.08** | 0.2±0.06 | 0.23±0.07 |
| | hard_expert | 0.65±0.01 | 0.66±0.05 | **0.67±0.04** | **0.67±0.02** | 0.52±0.1 | 0.35±0.17 |
| | hard_mixed | 0.32±0.14 | **0.34±0.11** | 0.3±0.1 | **0.34±0.08** | 0.23±0.08 | 0.16±0.17 |
| **Generalization** | norm_general | 0.34±0.05 | 0.35±0.04 | 0.29±0.09 | **0.37±0.04** | 0.29±0.06 | 0.09±0.1 |
| | hard_general | 0.28±0.03 | 0.3±0.05 | 0.28±0.04 | **0.31±0.09** | 0.14±0.06 | 0.13±0.04 |
| | norm_hero_general | 0.17±0.03 | 0.13±0.02 | 0.14±0.04 | **0.2±0.09** | **0.2±0.06** | 0.13±0.04 |
| | hard_hero_general | 0.16±0.05 | **0.19±0.05** | 0.17±0.02 | 0.17±0.02 | 0.07±0.05 | 0.08±0.03 |
| | norm_oppo_general | **0.21±0.01** | 0.14±0.03 | 0.14±0.04 | 0.18±0.06 | 0.13±0.07 | 0.12±0.03 |
| | hard_oppo_general | **0.09±0.06** | 0.08±0.02 | **0.09±0.02** | 0.08±0.04 | 0.04±0.02 | 0.04±0.01 |
| **Multi-Task** | norm_multi_level | 0.43±0.09 | 0.36±0.02 | 0.34±0.04 | **0.44±0.02** | 0.38±0.11 | 0.22±0.07 |
| | hard_multi_level | **0.38±0.08** | 0.33±0.08 | 0.29±0.07 | 0.37±0.05 | 0.27±0.05 | 0.2±0.01 |
| | norm_multi_hero | 0.57±0.07 | 0.31±0.2 | 0.3±0.07 | **0.59±0.05** | 0.51±0.17 | 0.39±0.06 |
| | norm_multi_oppo | 0.09±0.04 | 0.08±0.05 | 0.07±0.03 | **0.12±0.04** | 0.07±0.03 | 0.02±0.01 |
| | norm_multi_hero_oppo | 0.3±0.04 | 0.23±0.1 | 0.26±0.07 | **0.31±0.07** | 0.26±0.03 | 0.07±0.02 |
| **Heterogeneous** | norm_stupid_partner | 0.11±0.15 | **0.33±0.06** | 0.24±0.17 | 0.22±0.14 | 0.16±0.09 | 0.08±0.05 |
| | norm_expert_partner | 0.36±0.09 | 0.52±0.1 | **0.57±0.04** | 0.55±0.22 | 0.31±0.15 | 0.07±0.06 |
| | norm_mixed_partner | 0.49±0.2 | **0.59±0.19** | 0.32±0.03 | 0.42±0.38 | 0.17±0.27 | 0.15±0.04 |
| **Sub-Task** | gain_gold_medium | 0.13±0.01 | 0.12±0.01 | 0.12±0.01 | **0.15±0.01** | 0.13±0.03 | 0.14±0.01 |
| | gain_gold_expert | 1.01±0.03 | 0.98±0.02 | 1.00±0.01 | **1.03±0.01** | 0.98±0.08 | 0.79±0.06 |
| | gain_gold_mixed | **0.64±0.29** | 0.41±0.21 | 0.41±0.1 | 0.46±0.16 | 0.25±0.23 | 0.13±0.04 |

- **Heterogeneous:** As expected, the presence of a low-ability partner can disrupt cooperation and hinder offline learning on the *stupid_partner* datasets, whereas an expert partner has the opposite effect, highlighting the limitations of existing research on heterogeneous offline MARL.

- **Sub-Task:** The offline baselines exhibit robust performance in the *Sub-Task* at a low training cost. Additionally, BC demonstrates a competitive capability as well.

- **Ablations of learning paradigms:** We conduct ablation experiments to investigate the impact of communication and the CTDE paradigm. Specifically, from the comparison of COMM-CQL and IND-CQL, we can reveal that incorporating communication generally results in better performance due to the promotion of cooperation. Surprisingly, we found that the independent paradigm (IND-ICQ) outperformed the CTDE paradigm (MAICQ), which may be attributed to the challenges in the CTDE paradigm associated with credit assignment of agents with distinct rewards and roles.

## 6 Conclusion

In this paper, taking into account the limitations of existing offline RL datasets about practical applications, we introduce `Hokoff`, based on Honor of Kings, a well-known MOBA game that offers a high level of complexity for simulating real-world scenarios. We present a comprehensive framework for conducting research in offline RL and release a diverse and extensive collection of datasets, incorporating various levels of difficulty and a range of research factors. Moreover, the chosen tasks for dataset collection not only cater to Offline RL but also serve the purpose of offline MARL. We replicate multiple offline RL and offline MARL algorithms and thoroughly validate these baselines on our datasets. The obtained results highlight the shortcomings of existing Offline RL methods, underscoring the necessity for further research in areas such as challenging task settings, generalization capabilities, and multi-task learning. All components, including the framework, datasets, and baseline implementations, discussed in this paper are fully open-source.

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

# A    Author Statement

The authors of this work would like to state that we bear full responsibility for any potential violation of rights, including copyright infringement or unauthorized use of data. We affirm our commitment to conducting this research in accordance with ethical guidelines and legal requirements.

We further guarantee that we will ensure access to the data[4] and the framework code[5] used in this study, making them available to interested researchers for verification and replication purposes. Additionally, we are committed to providing the necessary maintenance and support to ensure the longevity and accessibility of the data. For datasets, we have plans to consistently offer more datasets in the future. These datasets will include larger sizes for larger models, higher levels for expert agents, and novel design factors for other research directions. Updating our datasets is an ongoing and long-term effort, and we welcome contributions from the community. Regarding benchmarks, we will actively monitor the latest state-of-the-art (SOTA) algorithms in the offline RL domain and integrate them into our benchmarks. Additionally, we will develop new algorithms within the benchmarks based on existing datasets and baselines. This ensures that our benchmarks remain up-to-date and reflect the advancements in offline RL research.

Should any concerns or inquiries arise regarding the contents of this work or the associated data, we encourage readers and fellow researchers to contact us directly. We are dedicated to addressing any issues promptly and transparently to uphold the integrity of our research.

# B    Limitations and Future Works

In our future endeavors, we plan to integrate our framework with a large-scale deep reinforcement learning platform namely *KaiwuDRL*, specifically designed to support Honor of Kings. By doing so, we will gain access to greater computational resources, enabling us to delve deeper into our current research endeavors and expand our investigations.

# C    Additional Datasets Details

## C.1    HoK1v1

In the *Generalization* category, "norm_general" and "hard_general," have their corresponding datasets. For example, to sample the "norm_general" dataset, we let the level-1 model fight with level-0, level-2, and level-4 models. However, during the test stage, we assess the generalization capabilities of the trained model by letting it fight against the level-1 model. Details about how we sample the *generalization* datasets can be referred to Table. 6. The latter four experiments do not require extra datasets. For example, in the "norm_hero_general" experiment, we directly use the model trained on the "norm_medium" dataset and let the model control different heroes. This is possible because the "norm_medium" dataset only contains the fixed default hero "luban." Therefore, we use the model trained on this dataset to test its generalization ability at controlling different heroes.

In the *Sub-Task: Destroy Turret* category presented in Table 3, there are three datasets sampled, each consisting of 100 trajectories. Notably, these datasets lack an opponent hero, making them simpler in nature. This design choice allows for broad applicability, diversity, and cost-effectiveness in research endeavors.

The primary objective in the *Sub-Task: Destroy Turret* scenarios is to efficiently dismantle the enemy's turret and crystal, with the enemy hero removed. Consequently, we adopt the number of game frames elapsed from the start of the game until the crystal's destruction as our evaluation protocol. Equation 1 outlines the scoring methodology employed, following a similar approach as presented in [6]. The score is normalized by two factors: *random_frame_length*, set to 2880, and *expert_frame_length*, set to 1812. A higher score is achieved by minimizing the time required to destroy the crystal.

In addition, we have generated violin charts to represent the distribution of episode returns in each dataset as shown in Fig. 3. We calculate episode returns using the formula $R = \sum_{t=0}^{T} \gamma^t r^t$, where

---

[4]`https://sites.google.com/view/hok-offline`
[5]`https://github.com/tencent-ailab/hokoff`

gamma is set to 1.0 to showcase the overall rewards obtained throughout an entire episode. For the *Sub-Task: Destroy Turret* datasets of HoK1v1, we have normalized the scores based on Equation 1. The violin charts demonstrate the diverse distribution of episode returns within our datasets.

$$normalized\_sub\_task\_score = \frac{random\_frame\_length - frame\_length}{random\_frame\_length - expert\_frame\_length} \quad (1)$$

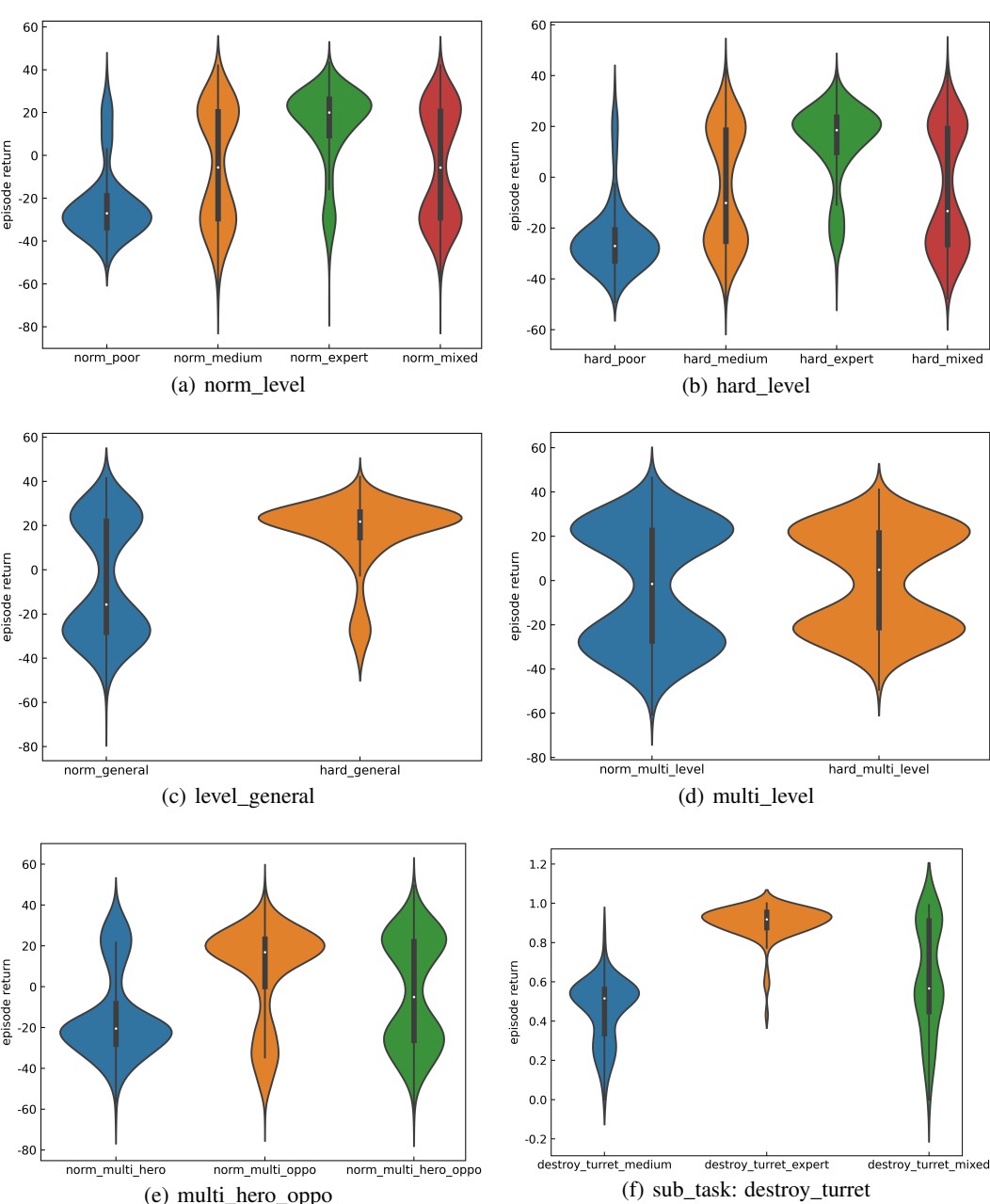

Figure 3: Violin diagrams of all datasets in HoK1v1.

## C.2 HoK3v3

The design of datasets and experiments pertaining to the concept of *Generalization* closely resembles that of the HoK1v1.

We introduce a sub-task called "*Gain Gold*" that builds upon the HoK3v3 game. In this modified version, we remove opponents and redefine the primary objective to focus on collecting gold within a limited number of time steps. This transforms the original competitive task into a resource collection task. Specifically, we set a maximum episode length of 8000 frames, and the controlled heroes are required to efficiently gather gold by killing monsters or creeps. As demonstrated in Table 3, we generate three datasets based on this sub-task, each consisting of 100 trajectories. The *gain_gold_medium* dataset is collected by a model with moderate performance, averaging 5904 gold collected. While, the *gain_gold_expert* dataset is obtained from a model with expert performance, averaging 12271 gold collected. Lastly, the *gain_gold_mixed* dataset combines the data from the previous two datasets equally. The scores are normalized based on Equation 2, where the $random\_gain\_gold$ is 5000 and $expert\_gain\_gold$ is 12000.

We have also generated violin charts in HoK3v3 to represent the distribution of episode returns in each dataset as shown in Fig. 4. The plot method used is similar to that in HoK1v1, with the exception of not using normalized scores in the *Sub-Task: Gain Gold*.

$$normalized\_sub\_task\_score = \frac{gain\_gold - random\_gain\_gold}{expert\_gain\_gold - random\_gain\_gold} \tag{2}$$

Table 6: Details of sampling *generalization* datasets.

| Environments | Datasets | Sampling and Training | | Testing |
| --- | --- | --- | --- | --- |
| | | Controlled side model | Opponent side model | Opponent side model |
| HoK1v1 | norm_general | level-1 | level-0,2,4 | level-1 |
| | hard_general | level-5 | level-0,2,4 | level-5 |
| HoK3v3 | norm_general | level-5 | level-1,4,7 | level-5 |
| | hard_general | level-8 | level-1,4,7 | level-8 |

# D   Environment Details

## D.1   Honor of Kings Arena

For a more detailed account of the game settings, please refer to the original paper [38] and its documentation[6] of Honor of Kings Arena. In this context, we will only summarize the critical information that is relevant to the RL research.

● **Observation Space**

We have utilized the fundamental set of observations presented in the aforementioned paper [38]. Specifically, the observation space of Honor of Kings Arena consists of a normalized vector with 725 dimensions, which includes five main components: *hero_state_common_feature*, *hero_private_feature*, *creep_feature*, *turret_feature*, and *global_feature*. The details of the observation vector are demonstrated in Table 7. In the table, *Main_camp* and *Enemy_camp* refer to the information of the controlled side and enemy side, respectively. Moreover, the information of invisible units is set to the default value.

● **Action Space** To tackle the complicated control, the Honor of Kings adopt a structured action space. Specifically, illustrated in Fig. 5 the action space is 6 dimensions, consisting of a triplet form, i.e. the action button, the movement or skill offset and the target game unit, which covers all the possible actions of the hero hierarchically: 1) what action button to take, e.g. skill or move.; 2) who to target, e.g., a turret, an enemy hero, or a creep in the troop; 3) how to act, e.g., the discretized direction to move and release skills [38]. Please refer to Table 8 for details of action space in HoK1V1.

● **Action Mask** There are two action masks designed to reduce the complexity of the action space, namely the *legal_action_mask* and the *sub_action_mask*. The former is constructed based on the rules of the game in order to exclude illegal actions, while the latter is determined by the selected button to eliminate actions that cannot be executed simultaneously with the chosen button, such as 'skill offset' and 'target unit' are not needed for 'move'.

---

[6]`https://aiarena.tencent.com/hok/doc/`

Table 7: Details of observation vector in Honor of Kings Arena

| feature name | dimensions | description |
| --- | --- | --- |
| Main_camp_hero_state_common_feature | 102 | hero's status, including whether it's alive, its ID and its health points (HP) |
| Main_camp_hero_private_feature | 133 | hero's specific kill information |
| Enemy_camp_hero_state_common_feature | 102 | enemy hero's status, including whether it's alive, its ID, its health points (HP) |
| Enemy_camp_hero_private_feature | 133 | enemy hero's specific kill information |
| Public_feature | 14 | visible information because of the turret |
| Main_camp_soldier_feature | 18*4 | the status of the creeps in a troop, including location and HP |
| Enemy_camp_soldier_feature | 18*4 | the status of the enemy's creeps in a troop, including location and HP |
| Main_camp_organ_feature | 18*2 | the status of turret and crystal |
| Enemy_camp_organ_feature | 18*2 | the status of enemy's turret and crystal |
| Global_feature | 25 | the period of the match |

Table 8: Description of action space in HoK1v1

| Action Class | Numbers | Description |
| --- | --- | --- |
| Button | 12 | what action button to take, e.g. skill or move. |
| Move X | 16 | move direction along X-axis. |
| Move Y | 16 | move direction along Y-axis. |
| Skill X | 16 | skill offset along X-axis. |
| Skill X | 16 | skill offset along Y-axis. |
| Target | 8 | who to target, e.g., a turret or an enemy hero |

• **Reward Design** The basic hero reward is a weighted average of several reward items, which is demonstrated in Equation 3. Subsequently, the hero's reward is transformed into a zero_sum value by subtracting the enemy's reward from it, as shown in Equation 4. Here, *team_reward* represents the average reward of the heroes within the team. Details of the reward items are demonstrated in Table 9.

.

$$
\begin{aligned}
hero\_reward =& w_1 * farming\_related + w_2 * KDA\_related + w_3 * damage\_related \\
& + w_4 * pushing\_related + w_5 * win/lose\_related
\end{aligned}
\tag{3}
$$

$$
hero\_reward_{zero\_sum} = hero\_reward - team\_reward_{enemy}
\tag{4}
$$

## D.2 Honor of Kings 3v3 Arena

For a more detailed account of the game settings, please refer to the documentation of Honor of Kings 3v3 Arena (HoK3v3) [7]. The environment code of HoK3v3 is integrated into the open-source 1v1 code, both with official authorization from Honor of Kings [8].

• **Observation Space** Specifically, the observation space of HoK3v3 consists of a normalized vector with 4586 dimensions. The details of observation vector are presented in Table 10.

• **Action Space** The form of action space in HoK3v3 is similar to that in HoK1v1 while the number of actions is larger. Description of action space in HoK3v3 is presented in Table 11.

---

[7] https://doc.aiarena.tencent.com/paper/hok3v3/latest/hok3v3_env/honor-of-kings/
[8] https://github.com/tencent-ailab/hok_env

Table 9: Description of reward items in HoK1v1

| Items | Type | Description |
|---|---|---|
| hp_point | dense | the rate of health point of hero |
| tower_hp_point | dense | the rate of health point of tower |
| money | dense | the increment of gold |
| ep_rate | dense | the rate of mana point |
| death | sparse | being killed |
| kill | sparse | killing an enemy hero |
| exp | dense | the increment of experience |
| last_hit | sparse | the lst hit for soldier |

Table 10: Details of observation vector in HoK3v3.

| feature name | dimensions | description |
|---|---|---|
| FeatureImgLikeMg | 6*17*17 | image-like feature, comprising six channels, which include barriers, grass, and other elements. |
| VecFeatureHero | 6*251 | the status of six heroes from the respective of controlled hero. |
| MainHeroFeature | 44 | private information of controlled hero. |
| VecSoldier | 20*25 | the status of all creeps. |
| VecOrgan | 6*29 | the status of turrets and crystals in both side. |
| VecMonster | 20*28 | the status of all monsters. |
| VecCampsWholeInfo | 68 | the status feature of the whole game. |

• **Reward Design** The basic hero reward is a weighted average of several reward items. Then the reward of each hero is processed to be zero_sum in minus the team reward of enemy which is the average of the hero rewards of 3 enemy heroes. The details of reward items are demonstrated in Table 12

## E    Framework APIs

We provide an example of the APIs in our framework, Listing 1. A comprehensive account of our framework can be found in our readily accessible open-access code repository.

## F    Evalution Protocols: Multi-Level Models

Based on the parallel training system named **SAIL** proposed by previous work [45], we have extracted and published several checkpoints from pre-trained dual-clip PPO [45, 44] models with varying levels determined by the outcome of the battle separately for HoK1v1 and HoK3v3.

Here, we present tables displaying the win rate of each level model against the model listed directly below it. The win rate is calculated with fixed hero selection, i.e. *luban* for HoK1v1 and

Table 11: Description of action space in HoK3v3

| Action Class | Numbers | Description |
|---|---|---|
| Button | 13 | what action button to take, e.g. skill or move. |
| Move | 25 | move direction. |
| Skill X | 42 | skill offset along X-axis. |
| Skill X | 42 | skill offset along Y-axis. |
| Target | 39 | who to target, e.g., a turret or an enemy hero |

Table 12: Description of reward items in HoK3v3

| Items | Type | Description |
|---|---|---|
| hp_rate_sqrt_sqrt | dense | the fourth root of the rate of health point of hero |
| money | dense | the increment of gold |
| exp | dense | the increment of experience |
| tower | dense | the rate of health point of turrets |
| killCnt | sparse | kill an enemy |
| assistCnt | sparse | assisting in the termination of an adversary |
| deadCnt | sparse | being killed |
| total_hurt_to_hero | dense | damage dealt to the enemies |
| atk_monster | dense | attack an monster |
| atk_crystal | dense | attack the crystal of enemy |
| win_crystal | sparse | destroy the crystal of enemy |

*{zhaoyun,diaochan,liyuanfang}* for HoK3v3, which may not right for other hero selection. Table 13 presents the win rate in the HoK1v1, where the *win_rate* column represents the win rate of *model1* against *model2*. Table 14 displays the win rate in HoK3v3. Additionally, we have included an API in our framework that allows researchers to conveniently test the win rate between any levels.

Table 13: Win rate of multi-level models in HoK1v1

| model1 | model2 | win_rate |
|---|---|---|
| 1v1_level_1 | 1v1_level_0 | 88% |
| 1v1_level_2 | 1v1_level_1 | 79% |
| 1v1_level_3 | 1v1_level_2 | 59% |
| 1v1_level_4 | 1v1_level_3 | 97% |
| 1v1_level_5 | 1v1_level_4 | 70% |
| 1v1_level_6 | 1v1_level_5 | 73% |
| 1v1_level_7 | 1v1_level_6 | 70% |

Table 14: Win rate of multi-level models in HoK3v3

| model1 | model2 | win_rate |
|---|---|---|
| 3v3_level_1 | 3v3_level_0 | 97% |
| 3v3_level_2 | 3v3_level_1 | 83% |
| 3v3_level_3 | 3v3_level_2 | 50% |
| 3v3_level_4 | 3v3_level_3 | 65% |
| 3v3_level_5 | 3v3_level_4 | 63% |
| 3v3_level_6 | 3v3_level_5 | 59% |
| 3v3_level_7 | 3v3_level_6 | 80% |
| 3v3_level_8 | 3v3_level_7 | 82% |

# G   Additional Experimental Details

## G.1   Additional Algorithm Details

• **Encoder**: Due to the complexity of the observation space, it is necessary to utilize a well-designed encoder for effective feature extraction. Taking inspiration from the "divide and conquer" approach employed in previous works [45, 44], in each algorithm, we implement a shared encoder network to process features, instead of directly feeding raw observations into the policy or critic network. For further details on the design of the encoder network, please refer to the mentioned papers [45, 44] as well as our code.

• **BC**: Behavior clone with maximum likelihood estimation loss. While, in multi-agent setting, HoK3v3, we adopt shared parameter and independent learning paradigm [34].

- **CQL** [19]: The implementation of Conservative Q-Learning is based on the original version [19] designed for discrete action spaces[9].

- **QMIX+CQL**: Due to the decoupling of control dependencies [45], the action space of HoK is structured with multi-head, which is similar to the joint action space in multi-agent settings. Inspired by this, we propose QMIX-CQL by incorporating mixer in QMIX [29] with CQL and use global Q to calculate td error term and use local Q to calculate CQL-loss term.

- **TD3+BC** [7]: Our implementation of TD3-BC is based on the open-source code[10]. In addition, the policy network and critic network share an encoder, which is updated simultaneously by both losses. Besides, We utilize Gumbel-Softmax reparameterization method to generate discrete actions for TD3 [8].

- **IQL** [16]: We implement IQL based on the open-source pytorch version[11]. The network design is similar to TD3-BC except for an additional value network.

- **IND+CQL** and **COMM+CQL**: To accommodate a multi-agent setting, based on the implementation of CQL, we adopt the independent learning paradigm and shared parameters, referred to as IND-CQL. Additionally, we introduce COMM-CQL which adds communication between agents by means of shared information that is constructed using max pooling.

- **IND+ICQ** and **MAICQ** [43]: We implement IND+ICQ and MAICQ based on the original published code[12]. IND+ICQ adopts independent learning paradigm, while MAICQ adopts CTDE paradigm by decomposing the joint-policy under the implicit constraint. The actor and critic networks update the shared encoder simultaneously as TD3-BC.

- **OMAR** [28]: The open-access code[13] of OMAR is not suitable for a discrete action space. Consequently, based on the core idea of it, we have undertaken the task of re-implementing OMAR to accommodate a discrete version.

### G.2 Hyperparameters

We have compiled the hyperparameters of HoK1v1 and HoK3v3 in Tables 15 and 16, respectively. These tables encompass the parameters of the training process, algorithm and optimizer settings.

Regarding the computing resources employed in HoK1v1, we utilize the Tesla T4 GPU and the AMD EPYC 7K62 48-Core Processor CPU. For the sampling process, 50 CPU cores are utilized, and each dataset required approximately 30 to 40 minutes for sampling. During the training process, each training experiment is conducted with one Tesla T4 GPU and two CPU cores, with an average training time of 9 hours per seed for 500000 training steps.

Regarding the computing resources employed in HoK3v3, we utilize the Tesla T4 GPU and the AMD EPYC 7K62 48-Core Processor CPU. For the sampling process, 50 CPU cores are utilized, and each dataset required approximately 80-90 minutes for sampling. During the training process, each training experiment is conducted with one Tesla T4 GPU and four CPU cores, with an average training time of 20 hours per seed for 500000 training steps.

Consequently, a total of 14 GPUs and 552 CPU cores are used to accommodate the overall computation requirements.

### G.3 Additional Results Discussion

- **Why is the performance of baseline models in the HoK1v1 comparatively inferior to those in the HoK3v3 setting?**

The experimental results conducted in the HoK1v1 reveal that the performance of baseline models is comparatively inferior to those in the HoK3v3 setting. This disparity can be attributed to the higher level of adversarial conditions present in the HoK1v1 environment. Furthermore, within the context

---

[9]https://github.com/aviralkumar2907/CQL/tree/master/atari

[10]https://github.com/sfujim/TD3_BC

[11]https://github.com/gwthomas/IQL-PyTorch

[12]https://github.com/YiqinYang/ICQ/tree/5a4da859ef597005040f79128ee6163547cf178d

[13]https://github.com/ling-pan/OMAR

Table 15: Hyperparameters for HoK1v1. The values of hyperparameters for algorithms are derived from their original implementation.

| Hyperparameters | Value |
| --- | --- |
| Batch Size | 128 |
| $\gamma$ | 0.99 |
| Max Steps (Exclude *Sub-Task* Datasets) | 500000 |
| Max Steps (*Sub-Task* Datasets) | 100000 |
| LSTM Time Steps | 16 |
| $\tau$ (Soft-Target-Update) | 0.005 |
| num_threads | 2 |
| final_evaluation_episodes | 150 |
| CQL $\alpha$ | 10.0 |
| TD3+BC $\alpha$ | 2.5 |
| IQL $\tau$ | 0.7 |
| IQL $\beta$ | 3.0 |
| Optimizer | Adam |
| beta1 | 0.9 |
| beta2 | 0.999 |
| eps | 1.00E-08 |
| Learning Rate | 3.00E-04 |

Table 16: Hyperparameters for HoK3v3. The values of hyperparameters for algorithms are derived from their original implementation.

| Hyperparameters | Value |
| --- | --- |
| Batch Size | 512 |
| $\gamma$ | 0.99 |
| Hard Update Frequency | 2000 |
| Max Steps | 500000 |
| Max Steps (*Sub-Task*) | 100000 |
| Iteration Steps | 1000 |
| Buffer Workers | 2 |
| num_threads | 4 |
| final_evaluation_episodes | 150 |
| CQL $\alpha$ | 10.0 |
| ICQ critic $\beta$ | 1000 |
| ICQ policy $\beta$ | 0.1 |
| OMAR coe | 0.5 |
| Optimizer | Adam |
| beta1 | 0.9 |
| beta2 | 0.999 |
| eps | 1.00E-08 |
| Learning Rate | 1.00E-04 |

of HoK3v3, if one teammate makes a sacrifice during a battle, the remaining two teammates are able to maintain their collaboration and continue to fight. This aspect ensures a greater level of robustness in the HoK3v3 when compared to the HoK1v1.

- **What are the reasons behind the underperformance of TD3+BC and OMAR?**

TD3+BC and OMAR demonstrated subpar performance in HoK1v1 and HoK3v3, respectively. The main cause of their lackluster outcomes stems from the fact that TD3 [8], upon which TD3+BC and OMAR are built, is incompatible with discrete action spaces. To enhance OMAR's performance, we replaced TD3 with advantage-weighted BC. This modification resulted in performance improvements.

- **QMIX+CQL in HoK3v3.**

We also implemented **QMIX+CQL** in the HoK3v3 game mode by adopting an independent learning paradigm. We thoroughly validated the performance of **QMIX+CQL** and compared it with **IND+BC** and **IND+CQL**, aggregating the results in Table 17. It is demonstrated that **QMIX+CQL** exhibits superior performance compared to **IND+CQL**, indicating that our novel method is also suitable for multi-agent settings with a structured action space.

Table 17: Validation of **QMIX+CQL** in HoK3v3 game mode.

| Factors | Datasets | IND+BC | IND+CQL | QMIX+CQL |
|---------|----------|--------|---------|----------|
| **Multi-Level** | **norm_poor** | 0.1±0.01 | 0.03±0.01 | **0.11±0.03** |
| | **norm_medium** | 0.48±0.01 | 0.4±0.03 | **0.52±0.04** |
| | **norm_expert** | 0.52±0.03 | 0.84±0.06 | **0.85±0.04** |
| | **norm_mixed** | 0.35±0.25 | 0.46±0.12 | **0.47±0.29** |
| | **hard_poor** | **0.16±0.03** | 0.12±0.03 | 0.13±0.02 |
| | **hard_medium** | **0.38±0.05** | 0.31±0.02 | 0.37±0.08 |
| | **hard_expert** | 0.65±0.01 | 0.67±0.04 | **0.7±0.03** |
| | **hard_mixed** | 0.32±0.14 | 0.3±0.1 | **0.43±0.04** |
| **Generalization** | **norm_general** | **0.34±0.05** | 0.29±0.09 | **0.34±0.02** |
| | **hard_general** | 0.28±0.03 | 0.28±0.04 | **0.32±0.01** |
| | **norm_multi_hero_general** | 0.17±0.03 | 0.14±0.04 | **0.18±0.06** |
| | **hard_multi_hero_general** | 0.16±0.05 | 0.17±0.02 | **0.19±0.02** |
| | **norm_multi_oppo_general** | **0.21±0.01** | 0.14±0.04 | 0.14±0.03 |
| | **hard_multi_oppo_general** | 0.09±0.06 | 0.09±0.02 | **0.1±0.02** |
| **Multi-Task** | **norm_multi_level** | 0.43±0.09 | 0.34±0.04 | **0.45±0.02** |
| | **hard_multi_level** | **0.38±0.08** | 0.29±0.07 | 0.35±0.04 |
| | **norm_multi_hero** | **0.57±0.07** | 0.3±0.07 | 0.56±0.13 |
| | **norm_multi_oppo** | **0.09±0.04** | 0.07±0.03 | **0.09±0.02** |
| | **norm_multi_hero_oppo** | **0.3±0.04** | 0.26±0.07 | **0.3±0.03** |
| **Heterogeneous** | **norm_stupid_partner** | 0.11±0.15 | 0.24±0.17 | **0.55±0.05** |
| | **norm_expert_partner** | 0.36±0.09 | 0.57±0.04 | **0.72±0.07** |
| | **norm_mixed_partner** | 0.49±0.2 | 0.32±0.03 | **0.66±0.05** |
| **Sub_Task** | **gain_gold_medium** | **0.13±0.01** | 0.12±0.01 | **0.13±0.02** |
| | **gain_gold_expert** | 1.01±0.03 | 1±0.01 | **1.02±0.01** |
| | **gain_gold_mixed** | **0.64±0.29** | 0.41±0.1 | 0.58±0.17 |

# H  Additional Discussion

## H.1  The significance of our design factors in the context of offline reinforcement learning

**Task difficulty**: Intuitively, the level of difficulty in the environment significantly impacts the performance of algorithms. However, previous researches only utilized one set of datasets with a uniform level of difficulty in the environment. Providing datasets with diverse difficulty can not only more comprehensively evaluate the ability of offline algorithms but also be more suited for real-world tasks like HoK, which are characterized by diverse levels of difficulty.

**Multi-task**: Combining offline reinforcement learning with multi-task learning enables efficient use of limited data. Sharing knowledge[1] and representations across tasks enhances data efficiency, leading to more general and robust feature learning. Besides, multi-task learning facilitates knowledge transfer between tasks. Leveraging shared parameters and representations accelerates learning for the target task in offline reinforcement learning, benefiting from related tasks' knowledge.

**Generalization**: Firstly, in offline RL, learning is based on a fixed dataset collected from previous experiences. This dataset might not cover all possible scenarios, so the learned policy needs to generalize well to new, unseen situations to perform effectively. Secondly, real-world environments are often complex and diverse. A policy only limited to the dataset without generalizing would likely fail when facing even slightly different conditions. Generalization ensures the policy's adaptability to various situations in the real-world scenarios.

**Heterogeneous Teammate**: Heterogeneous teammates are a crucial research direction in Multi-Agent Reinforcement Learning (MARL). In practical scenarios such as HoK or other multi-agent systems, players typically possess varying capacities. Consequently, the datasets collected from real-world scenarios consist of heterogeneous teammate data, necessitating the need for corresponding research in the offline MARL domain.

### H.2 From the perspective of the Honor of Kings game, why offline reinforcement learning is necessary and what potential limitations exist when compared to online reinforcement learning?

Training agents for the Honor of Kings game using offline reinforcement learning (RL) offers several advantages, including reduced training time, lower computation resource requirements, and better utilization of existing data resources. We compare the computational and time costs of online RL and offline RL in Table 18. It is evident that training an online agent from scratch to reach specific levels (level_5 for HoK1v1 and level_7 for HoK3v3) requires thousands of CPU cores and dozens of hours. On the other hand, training offline agents to reach same levels only requires a few CPUs and a shorter training time using pre-collected datasets. Additionally, there is a wealth of previously collected battle data that can be used for training offline RL agents. However, compared to online RL, it is important to note that offline RL in the HoK game heavily relies on large amounts of high-level battle data to train expert-level agents, which may be a potential limitation.

Table 18: The comparison of the computational and time costs between online RL and offline RL.

| Online/Offline | CPU cores | GPU cores | Performance | Training time (hours) |
|---|---|---|---|---|
| HoK1v1 (online) | 4000 | 2 | 1v1_level_5 | 60h |
| HoK1v1 (offline) | 2 | 1 | 1v1_level_5 | 9h |
| HoK3v3 (online) | 1000 | 1 | 3v3_level_7 | 97h |
| HoK3v3 (offline) | 4 | 1 | 3v3_level_7 | 20h |

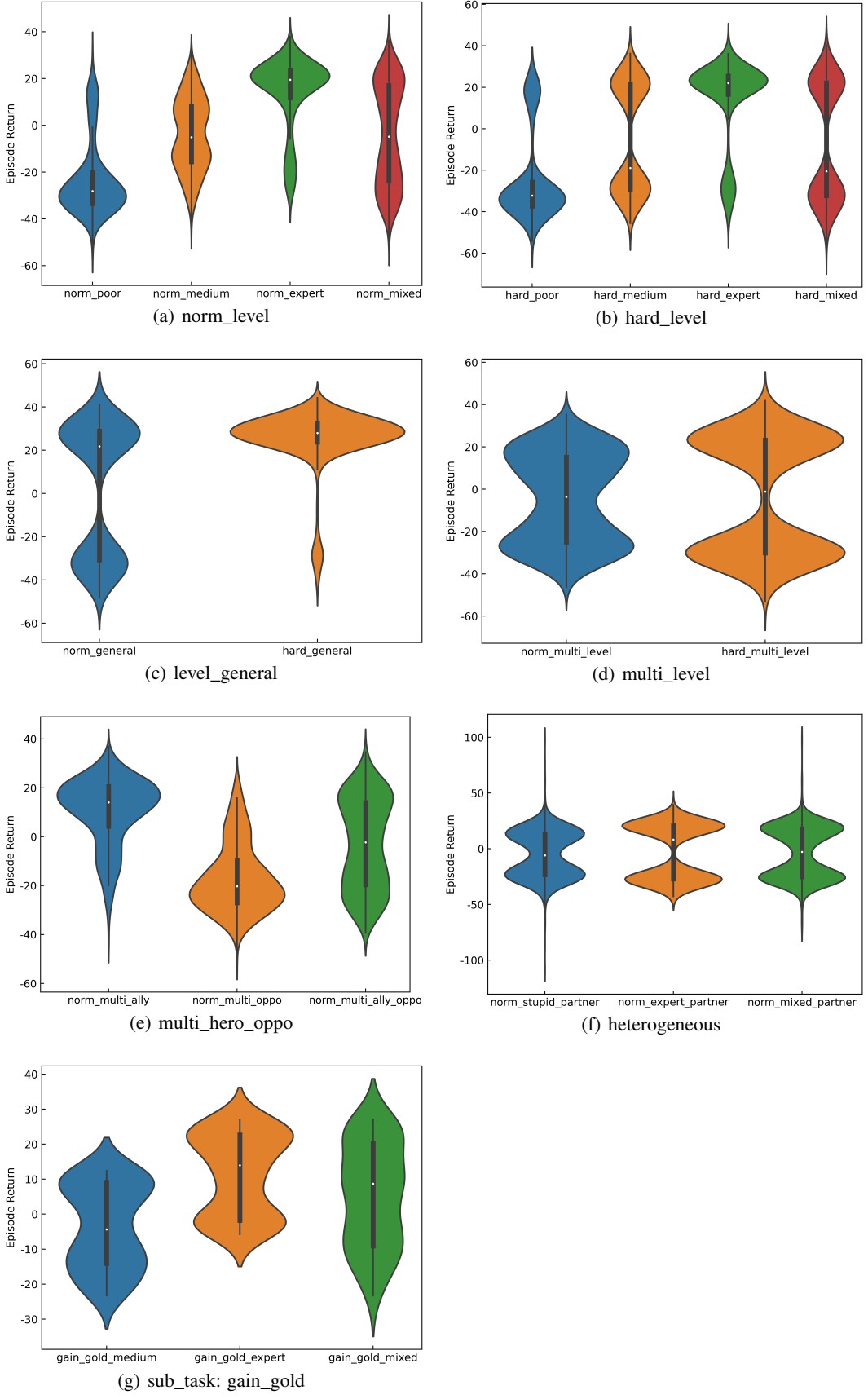

Figure 4: Violin diagrams of all datasets in HoK3v3.

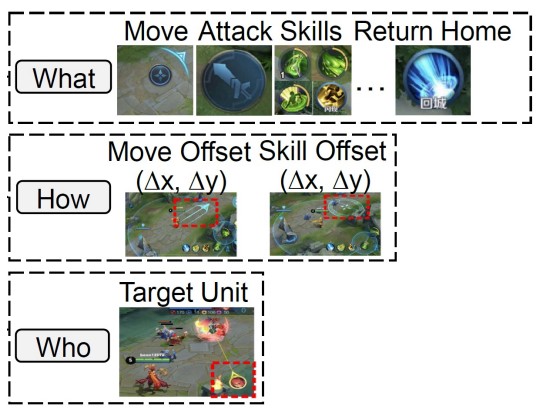

Figure 5: Action space in HoK1v1 [38]

```
1  cd   <root_path>
2
3  # sample example
4  sh offline_sample/scripts/start_sample.sh <args>
5
6  # train example
7  python offline_train/train.py --<args>
8
9  # evaluate example
10 python offline_eval/evaluation.py --<args>
11
```

Listing 1: APIs example

