# OpenReview forum: "Hokoff: Real Game Dataset from Honor of Kings and its Offline Reinforcement Learning Benchmarks"
_NeurIPS.cc/2023/Track/Datasets_and_Benchmarks — NeurIPS 2023 Datasets and Benchmarks Poster_

### Official Review · Reviewer_DxzA · 2023-07-19
**Proposing an offline reinforcement learning benchmark based on Honor of Kings with both dataset and benchmark, but more explanation and rationales are needed**

**Rating:** 6
**Confidence:** 3

**Strengths:**

1. The introduction of a benchmark for offline reinforcement learning and multi-agent reinforcement learning (MARL) based on the popular game Honor of Kings is a critical contribution for researchers in this community. This benchmark has the potential to accelerate the development of this field and aid in the evaluation of algorithms.

2. The data, code, and documentation provided in this work are comprehensive, making it easy for interested readers to get started with these materials. This level of detail is beneficial for researchers looking to build upon the proposed benchmark and apply it to their own work.

3. The paper is generally well-written and easy to follow, presenting the problem and proposed solution in a clear and concise manner. The organization of the paper and the use of appropriate technical language make it accessible to a wide range of readers, including both experts and newcomers to the field.


**Additional Feedback:**

None

**Clarity:**

The paper is well-written, but the authors could consider providing a more thorough explanation of the connection and difference between single-agent offline reinforcement learning and offline multi-agent reinforcement learning. This would help readers better understand the motivation for considering both sub-problems and the implications of the proposed benchmark. Providing a more detailed discussion of the relationships and differences between these two tasks would strengthen the paper's contribution to the field.

**Correctness:**

Overall, the construction of the dataset appears to be sound, and the evaluation methods and experimental design are generally appropriate and executed correctly.

**Documentation:**

The paper provides sufficient details on data collection, maintenance, and ethical considerations for the dataset. However, for the proposed benchmark, it would be valuable to include a maintenance plan. This plan could outline how the benchmark will be maintained and updated over time to ensure its continued relevance and usefulness.

**Ethics:**

No ethical issues have not identified.

**Limitations:**

The authors should consider discussing the potential limitations of their work. They could explore how and why the proposed benchmark may be useful for researchers in the field. This discussion could be divided into two parts:

1. From the perspective of offline reinforcement learning, the authors claim that the proposed benchmark is more complex than existing benchmarks like D4RL and RL Unplugged. However, to save computation, only simplified sub-tasks such as Honor of Kings Arena, Destroy Turret, and Gain Gold are implemented in this benchmark. The authors may provide a rationale for why these simplified tasks are sufficient and what potential limitations may arise from not including more complex scenarios.

2. From the perspective of the Honor of Kings game, the authors may explain why offline reinforcement learning is necessary and what potential limitations exist when compared to online reinforcement learning. This discussion would help readers understand the context in which the proposed benchmark is relevant and how it can contribute to advancing the field.

**Opportunities For Improvement:**

1. The paper addresses two sub-problems, namely single-agent offline reinforcement learning and multi-agent reinforcement learning. However, the authors do not provide a clear explanation of the connections or differences between these tasks, which may give readers the impression that the paper should be divided into two separate papers. Therefore, the authors should consider providing a more detailed discussion of the relationships and differences between these two tasks, especially in paragraphs 2 and 3.

2. The authors may want to add further explanation of why task complexity, generalization, multi-task learning, and similar considerations are crucial in the context of offline reinforcement learning. This would help readers better understand the motivations behind the proposed benchmark, and how it can contribute to the advancement of the field. Providing a more in-depth discussion of these concepts would strengthen the paper's contribution and significance.


**Relation To Prior Work:**

The authors reviewed the D4RL and RL Unplugged offline reinforcement learning benchmarks and highlighted that the main difference with their work is the inclusion of more complex tasks. It would be beneficial if the authors could expand on this discussion and consider the problem from different angles. For example, they could explore how the proposed benchmark addresses limitations of existing benchmarks, or how it allows for the evaluation of algorithms on tasks that are more representative of real-world scenarios. Additionally, the authors could discuss how the proposed benchmark can contribute to the development of more robust and generalizable offline reinforcement learning algorithms.

**Summary And Contributions:**

This paper focuses on the problem of offline reinforcement learning and introduces Hokoff, a novel dataset derived from the MOBA game Honor of Kings. The authors aim to address two aspects of offline reinforcement learning, namely single-agent and multi-agent reinforcement learning (MARL). For each sub-problem, the authors provide typical baselines to compare against. This benchmark has the potential to enhance existing offline RL methods in terms of task complexity, generalization, and multi-task learning. Overall, the introduction of the Hokoff dataset is an essential contribution to the field of offline reinforcement learning, offering a challenging and diverse set of tasks for researchers to evaluate their algorithms. The benchmark's potential to improve existing methods through promoting better generalization, task complexity, and multi-task learning is a promising aspect of this work.

---

> ### Author Response · Authors · 2023-08-10
>
> Thank you for your valuable feedback and suggestions for improving our paper!
>
> ●**Q1:  Explain the relationships and differences between offline single-agent reinforcement learning and multi-agent reinforcement learning.**
>
> Thank you for your reminder and suggestion. We have incorporated the connections and differences between offline Multi-Agent Reinforcement Learning and offline Reinforcement Learning into paragraph 4 of the Introduction.
> Here is the additional content:
>
> The connections between offline RL and offline MARL are closely linked due to similar challenges pertaining to offline learning. While, offline MARL also introduces distinct algorithm development requirements, as it involves unique characteristics like multiple agents and intra-team cooperation. The current offline MARL algorithms are mainly adaptations of offline RL algorithms, necessitating the availability of standardized offline datasets that cater to both single-agent and multi-agent settings. Thus, to enhance versatility and practicality, it is crucial to propose datasets that encompass both single-agent settings and multi-agent settings.
>
> ●**Q2: Why task complexity, generalization, multi-task learning, and similar considerations are crucial in the context of offline reinforcement learning**
>
> We have incorporated an additional discussion of the significance of our design factors in the context of offline reinforcement learning in Appendix H.1.
> Here is the brief summary:
>
> (1)Task difficulty: Providing datasets with diverse difficulty can not only more comprehensively evaluate the ability of offline algorithms but also be more suited for real-world tasks like HoK.
> (2)Multi-task: Combining offline reinforcement learning with multi-task learning enables efficient use of limited data and facilitates knowledge transfer between tasks.
> (3)Generalization: The fixed and limited datasets require the learned policy to generalize well to new, unseen situations to perform effectively.
> (4)Heterogeneous Teammate: The datasets collected from real-world scenarios consist of heterogeneous teammate data, necessitating the need for corresponding research in the offline MARL domain.
>
> ●**Q3: Why only simplified sub-tasks such as Honor of Kings Arena, Destroy Turret, and Gain Gold are implemented in this benchmark.**
>
> It is noteworthy that both Honor of Kings Arena (HoK1v1) and Honor of Kings 3v3 Arena (HoK3v3) employed in this benchmark are full MOBA games with no simplification. We have introduced two simplified sub-tasks, namely Destroy Turret and Gain Gold, derived from HoK1v1 and HoK3v3, respectively. These sub-tasks are designed based on the different sub-objectives present in the HoK games and require fewer resources for training.
>
> ●**Q4: From the perspective of the Honor of Kings game, the authors may explain why offline reinforcement learning is necessary and what potential limitations exist when compared to online reinforcement learning.**
>
> We have provided additional explanation for the necessity of offline reinforcement learning in the HoK game in Appendix H.2.
> Here is the brief summary:
>
> As evidenced by the data comparison in Appendix H.2, training agents for the Honor of Kings game using offline reinforcement learning (RL) offers several advantages, including reduced training time, lower computation resource requirements, and better utilization of existing data resources.
>
> However, compared to online RL, it is important to note that offline RL in the HoK game heavily relies on large amounts of high-level battle data to train expert-level agents, which may be a potential limitation.
>
> **Not finished yet. Please refer to the next block.**

---

> > ### Author Response · Authors · 2023-08-10
> >
> > ●**Q5: how the proposed benchmark addresses limitations of existing benchmarks, or how it allows for the evaluation of algorithms on tasks that are more representative of real-world scenarios?**
> >
> > As presented in our paper, there are several limitations of existing datasets and benchmarks. D4RL and RL Unplugged primarily focus on relatively simple tasks and lack high-dimensional, practical, and multi-agent tasks that closely resemble real-world scenarios. In contrast, our datasets and benchmarks are built upon Honor of Kings, which is a successful MOBA game with sufficient complexity, including a high-dimensional state-action space and multi-agent settings. Additionally, prior work [2] has demonstrated that the model trained based on the HoK environment can be transferred into the real game and battle with human players, which guarantees the real-world application potential of our benchmarks.
> >
> > StarCraft II Unplugged only utilizes a dataset derived from human replays, which lacks diversity in design for offline RL, and they did not evaluate existing offline RL methods. While, the design of our datasets fully considers the diversified needs of offline RL. Furthermore, we have thoroughly implemented and evaluated existing offline RL and offline MARL methods and introduced Multi-Level Models as an evaluation protocol to fully verify the performance of offline RL algorithms.
> >
> > ●**Q6: Additionally, the authors could discuss how the proposed benchmark can contribute to the development of more robust and generalizable offline reinforcement learning algorithms?**
> >
> > In fact, our dataset design prioritizes the development of robustness and generalization through factors such as "multi-difficulty," "multi-task," and "generalization." These aspects have often been overlooked by previous research works. The benchmarks we propose are user-friendly, facilitating the development of offline RL algorithms, while ensuring the evaluation of robustness and generalization. The experimental results highlight the limitations of existing methods in tasks related to robustness and generalization, thereby drawing attention to these areas.
> >
> > ●**Q7: Maintenance plan.**
> >
> > The maintenance plan for our datasets and benchmarks has been added in Appendix A.
> >
> > **Reference**
> >
> > [1]. Yu T, Kumar A, Chebotar Y, et al. Conservative data sharing for multi-task offline reinforcement learning[J]. Advances in Neural Information Processing Systems, 2021, 34: 11501-11516.
> >
> > [2]. Ye D, Liu Z, Sun M, et al. Mastering complex control in moba games with deep reinforcement learning[C]//Proceedings of the AAAI Conference on Artificial Intelligence. 2020, 34(04): 6672-6679.

---

> > > ### Comment · Reviewer_DxzA · 2023-08-13
> > >
> > > I have read the comments by the authors and checked the updated manuscript and appendix files. I am generally satisfied with the current version and would like to acknowledge the authors' efforts.

---

> > > > ### Author Response · Authors · 2023-08-15
> > > >
> > > > Thank you very much for dedicating your time to reviewing the revised manuscript and supplementary materials. Your feedback and guidance have been immensely valuable in shaping the current version of the paper. We are grateful for your thorough review and positive assessment of the paper. Your insights have greatly contributed to its overall improvement.

---

### Official Review · Reviewer_KDwt · 2023-07-20
**lack of diversity in the benchmark**

**Rating:** 6
**Confidence:** 2
**Correctness:** Appears correct

**Strengths:**

Really nice benchmark for someone developing algorithms for honor of kings. Therefore, this is a valuable benchmark for people developing for that specific game

**Additional Feedback:**

The paper needs a rewrite with additional examples and must justify why one game is enough as a benchmark. Otherwise, this is a hard sell.

**Clarity:**

Clear



**Documentation:**

Yes, the GitHub repository is detailed and provides the necessary details for implementation

**Limitations:**

Have not been discussed

**Opportunities For Improvement:**

I do not see any opportunity except for adding more games in it. However, that might be really large overhaul. An RL venue with many researchers developing specifically for HOK is more suitable for this paper.

Fig,2 is unreadable.

**Relation To Prior Work:**

Ample citations and prior discussion

**Summary And Contributions:**

This paper provides a benchmark for offline single agent and multi agent RL. Specifically, it allows users to leverage honor of kings in the 1V1 and 3V3 setup for RL research. They provide a total of five benchmarking methods for setup with this algorithm.

While it feels like Honor of kings is a difficult enough game to play using RL. For a researcher in RL to use a benchmark that just provides one game is really specific. Unless I build an RL algorithm for specifically this game, this will be rather cumbersome to do. Ideally, I would like to test my new algorithm on many many games both multi agent and single agent. No one nowadays accepts a paper with just one game, at least in top venues.

This is my major criticism, I think a benchmark provide more than one example otherwise adapting my algorithm for this would not be beneficial to the algorithm developer.

---

> ### Author Response · Authors · 2023-08-10
>
> Thank you for acknowledging the quality and value of our benchmark in your feedback. We sincerely appreciate your well-meaning concerns and are grateful for the opportunity to address them in this rebuttal response.
>
> ●**Q1:  Why one game is enough as a benchmark.**
>
> We understand your concern about the limited scope of a single game in providing sufficient diversity and facilitating the algorithm development. However, we can explain how this concern can be eliminated in terms of both game settings and algorithm development.
>
> Regarding game settings, as highlighted in our paper, the HoK game serves as an ideal platform due to its higher complexity and significant flexibility, resulting in a diverse set of scenarios. (1) Notably, HoK offers two distinct maps: HoK1v1, which focuses on single-agent competition, and HoK3v3, which emphasizes multi-agent cooperation and competition. These two maps differ significantly in terms of map design, units, hero roles, and battle strategies, thereby making them appear as two separate games that only share the MOBA battle logic and hero skill logic. (2) Furthermore, the two maps serve as foundations for designing various subtasks, as exemplified in the paper. These subtasks showcase the ease with which additional challenges can be formulated within the HoK framework. (3) Additionally, the game features numerous heroes, each possessing unique skills, further contributing to the diversity of battle tasks that can be undertaken. (4) In line with the demand for diversity in offline RL, we have meticulously designed several dataset-design factors that align with real-world applications of both HoK and other relevant scenarios. As a result, we have proposed a series of datasets in the paper that adequately address this requirement.
>
> In the realm of algorithm development, it is worth noting that the HoK environment, along with the datasets we have proposed, can serve as a valuable resource for the advancement of general reinforcement learning algorithms. Numerous studies have attested to its effectiveness and the absence of constraints[8,9,10]. Besides, creating specialized algorithms tailored for the HoK has also proven to be highly successful[3,4,5,6,7]. This can be attributed to the existence of real-world applications within the game.
>
> In addition, it is noticed that there are several noteworthy works accepted in top venues that focus on specific one game and make significant contributions to the research community. For instance, SMAC[1] has emerged as the paramount benchmark environment for Multi-Agent Reinforcement Learning (MARL) in the present day. Additionally, HoK Arena[2] introduced the HoK1v1 environment to the community for the first time and has just been accepted in the Datasets and Benchmarks Track at NeurIPS 2022.
>
> ●**Q2: Fig,2 is unreadable.**
>
> We apologize for any confusion resulting from the unclear Figure 2 in our paper. We have enhanced the visual representation of Figure 2 by repainting it. Additionally, we have improved the clarity and accuracy of the picture's caption.
>
> **Not finished yet. Please refer to the next block.**

---

> > ### Author Response · Authors · 2023-08-10
> >
> > **Reference:**
> >
> > [1]. Samvelyan M, Rashid T, Schroeder de Witt C, et al. The StarCraft Multi-Agent Challenge[C]//Proceedings of the 18th International Conference on Autonomous Agents and MultiAgent Systems. 2019: 2186-2188.
> >
> > [2]. Wei H, Chen J, Ji X, et al. Honor of kings arena: an environment for generalization in competitive reinforcement learning[J]. Advances in Neural Information Processing Systems, 2022, 35: 11881-11892.
> >
> > [3]. Wu B. Hierarchical macro strategy model for moba game ai[C]//Proceedings of the AAAI conference on artificial intelligence. 2019, 33(01): 1206-1213.
> >
> > [4]. Gao Y, Shi B, Du X, et al. Learning diverse policies in moba games via macro-goals[J]. Advances in Neural Information Processing Systems, 2021, 34: 16171-16182.
> >
> > [5]. Ye D, Liu Z, Sun M, et al. Mastering complex control in moba games with deep reinforcement learning[C]//Proceedings of the AAAI Conference on Artificial Intelligence. 2020, 34(04): 6672-6679.
> >
> > [6]. Ye D, Chen G, Zhang W, et al. Towards playing full moba games with deep reinforcement learning[J]. Advances in Neural Information Processing Systems, 2020, 33: 621-632.
> >
> > [7]. Ye D, Chen G, Zhao P, et al. Supervised learning achieves human-level performance in moba games: A case study of honor of kings[J]. IEEE Transactions on Neural Networks and Learning Systems, 2020, 33(3): 908-918.
> >
> > [8]. Wei H, Ye D, Liu Z, et al. Boosting Offline Reinforcement Learning with Residual Generative Modeling[C]//30th International Joint Conference on Artificial Intelligence, IJCAI 2021. International Joint Conferences on Artificial Intelligence, 2021: 3574-3580.
> >
> > [9]. Jiang D, Ekwedike E, Liu H. Feedback-based tree search for reinforcement learning[C]//International conference on machine learning. PMLR, 2018: 2284-2293.
> >
> > [10]. Eisenach C, Yang H, Liu J, et al. Marginal Policy Gradients: A Unified Family of Estimators for Bounded Action Spaces with Applications[C]//International Conference on Learning Representations. 2018.
> >
> > [11]. Wang Q, Xiong J, Han L, et al. Exponentially weighted imitation learning for batched historical data[J]. Advances in Neural Information Processing Systems, 2018, 31.

---

> ### Comment · Reviewer_KDwt · 2023-08-14
> **Thank you for your response**
>
> While I personally feel that RL algorithmic development on one particular environment is very unlikely to be robust enough to carry forward to other environments. specially because, the neural network is quite adept at zeroing on certain peculiarities on a particular environment.
>
> However, it seems like even having a benchmark for one game is a lot of work and extending a benchmark to multiple games and providing interfaces for them would be monumental level of work.  In light of this and since, my main argument is the lack of diversity,  I will raise my score to the average of the other two reviewers.
>
> I do not believe this is the right way for RL benchmarks but I recognize the amount of work that would go into getting these interfaces for multiple games setup.

---

> > ### Author Response · Authors · 2023-08-15
> >
> > We sincerely appreciate your dialectical viewpoint of this paper. Your thoughtful evaluation of our work and the concerns you raised regarding the robustness and generalizability of the RL algorithm in specific environments are highly appreciated. Your feedback helps us recognize the significance of diversity in environments when developing algorithms.
> >
> > We genuinely appreciate the time and effort you've dedicated to reviewing our paper, and your constructive critique will undoubtedly contribute to enhancing the quality of our research.

---

### Official Review · Reviewer_D3sf · 2023-07-21
**Needs some clarity improvements**

**Rating:** 7
**Confidence:** 3
**Correctness:** I believe the work presented is correct.

**Strengths:**

1. The idea seems sound; MOBA games seem like a reasonable source of complex, interesting behaviors for demonstrating new methods on.

1. The datasets presented span a variety of potential research directions (generalization, multi-agent work, etc), making their benchmark widely useful. Additionally, it seems like their method could be readily extended to even more settings, and harder tasks.

1. They provide initial results for a number of baseline methods, validating the design and utility of their framework, as well as providing a base on which others can build.

**Additional Feedback:**

Other than daily active users, I'm curious if there were other reasons for selecting Honor of Kings over other MOBAs. Not that I am objecting to its choice, merely curious if it provides mechanics/settings that are of more interest.

**Clarity:**

The paper's clarity could be improved, as I mentioned in Opportunities for Improvement. Parts are too vague (e.g. Section 4.1), some information is missing from the main body (e.g. dataset size), and some terminology ("tasks") seems to be used in different, conflicting ways. Additionally, I found Section 4.4 challenging to understand overall; the information seems a bit scattered and lacking in necessary detail to understand what is being presented. For the most part this is remedied by looking at the Appendix, however in some cases it only increased my confusion because the paper and the Appendix are in conflict.

**Documentation:**

The site for the dataset seems easy to use, and the code looks well-documented. A license is provided, but I did not see a hosting or maintenance plan.

**Limitations:**

I am not opposed to encouraging sample-efficient learning, but 1000 trajectories seems like a relatively small amount of data for such complex tasks. By comparison, it seems like the StarCraft II dataset you mention in the paper has 2.8 million total episodes. I understand that this is an area you seem to be working on, but I think it should at least be stated up-front in the main body of the paper. The results from your baselines look reasonable for a dataset in general, as there is reasonable success but also room for future work using your dataset to improve, but I wonder partially how much of that is due to the dataset size. (The problem could alternatively be framed and justified as encouraging sample-efficiency, but that is not currently being done.)

**Opportunities For Improvement:**

Overall I think the work has merit and would be valuable to the community, but there are significant issues with the current paper (for example indicated that there is a Gain Gold subtask when none such seems to be available). If they are corrected, I will raise my score.

1. There are some clarity issues (discussed below), so I may be misunderstanding, but it seems as though two of the multi-task settings are based on varying the levels of the opponent, as per:
> Thus, we propose several level-based multi-task datasets by sampling data with randomly selected opponent levels

    First, this does not align with what I think I and most others think of as "tasks" in the multi-task setting. I expect different tasks to involve learning meaningfully different skills. I think the multi-task setting that involves being against randomly selected opponents is reasonable, since different opponents might require different ways to counter them, but I'm not sure that level makes sense to me. My expectation upon going into that section was that it would have multiple of what the paper refers to as sub-tasks (destroy turret, gain gold), which I would also encourage you to consider. It is unclear to me that random opponent levels require different skills from the agent.

    Second, I am unsure how this is meaningfully distinct from the "mixed Multi-Difficulty" dataset proposed.

    Third, in the beginning of the paper, you say:
    > term "tasks" to refer to different maps within the game, such as HoK1v1 and HoK3v3.
    which both does not seem to align to your later usage of the term (as in multi-task), and "maps" (while probably true) seems like an odd/misleading way to describe HoK1v1 and HoK3v3 given that their biggest distinction is the fact that one is a 1v1 setting and the other is a 3v3 setting.

1. Some of the details provided in the Appendix should be provided in the main body of the paper: for example, the fact that most of the datasets have 1000 trajectories except the Destroy Turrets sub-task which has 100 trajectories, and the fact that a specific hero selected as the default (as it was not clear to me in the Multi-task description what the random hero selection was in contrast to).

1. The paper indicates that there are two sub-tasks (Gain Gold, Destroy Turrets). However I only see details in the paper for Destroy Turrets, and at the dataset website only Destroy Turrets is available to download. At minimum, "sub-tasks" should be changed to accurately reflect what is available. Ideally, it would be optimal to have more than one (or even two) "sub-tasks".

1. In the paper you say:
> we have identified three key challenges for generalization: hero generalization, opponent generalization, and level generalization. To facilitate the research on these challenges, we have developed corresponding datasets and test-tasks respectively, including datasets and test-tasks named norm_general and hard_general for level generalization, test-tasks named hero_general for hero generalization and test-tasks named oppo_general for opponent generalization

    but in the Appendix you say:

    > In the Generalization category, there are four experiments conducted without datasets, namely multi_ally_general and multi_oppo_general with norm or hard level. In these experiments, we directly evaluate the performance of the trained models, norm_medium and hard_medium

    These descriptions are inconsistent. Additionally, some of the detail provided in the Appendix here would be helpful in the main paper (i.e. explicitly stating that you conducted the experiments without datasets, rather than allowing that to be implied from "Generalization").

Clarity improvements:

1. Section 4.1 is lacking in detail. First, it indicates the framework is good at sampling diverse datasets, but does not specify why this is true. Second, it largely defers to Figure 2, with no in-text description of the figure. Third, the figure itself is unclear, largely due to the fact that it references "Multi-Level Models", which is not described until 2 pages later (in Section 4.3). At the very least there should be some reference (i.e. "Multi-Level Models are described in Section <X>") Fourth, Multi-Level Models in the figure seem to be used for both sampling and evaluation (which also aligns with my understanding of how you collected the datasets), but they are described in Section 4.3 as part of the evaluation procedure exclusively (then referenced briefly in 4.4). I think clarity could be improved by moving this key part of your framework up to earlier in the paper.

1. On line 158: Why are the observations imperfect? How imperfect?

1. Is level of difficulty *only* based on the level of the opponents? It's ambiguous from the wording. If it's not, what other factors are there?

1. Tables 3 and 4 in the Appendix were quite useful to me, for getting an overall impression of the datasets provided. I understand you don't have room in the main body for it, but even just an explicit reference ("A summary of our datasets is provided in the Appendix, in Tables 3 and 4") would have been useful.


Minor:

1. Lines 109-111 and 153-155 or so seem to be duplicates of each other.

**Relation To Prior Work:**

Yes, the paper outlines similar work and how this method is distinct. I have not done Offline RL in this space, so I am unsure if it is missing anything critical.

**Summary And Contributions:**

This work presents a set of Offline RL and MARL datasets for a variety of settings (such as to test generalization, multi-agent capability, etc) for the game Honor of Kings, as well as results on the benchmark for a number of baselines.

---

> ### Author Response · Authors · 2023-08-10
>
> We would like to express our sincere gratitude for your approval of our work, and we extend our heartfelt appreciation for your constructive feedback. We sincerely apologize for any confusion caused by certain clarity issues and honest mistakes and we are committed to addressing these shortcomings.
>
> ●**Q1: The confusing meaning of term "tasks" .(multi-task or map)**
>
> We apologize for the misalignment of the term "tasks" between the paper's initial presentation and its subsequent usage. As a remedy, we propose to use the term "game modes" instead of "tasks" to accurately represent the various settings within the game (e.g., 1v1 or 3v3). In order to maintain consistency, we have made revisions to all relevant sections of the paper.
>
> ●**Q2: There is no details in the paper for Gain Gold and corresponding datasets seems to be unavailable.**
>
> For detailed information regarding the "Gain Gold" sub-tasks, descriptions are available in the last two paragraphs in Appendix C.2. Additionally, Table 6 provides an in-depth presentation of the "Gain Gold" datasets. We apologize that there was an initial inaccuracy in the title of the "Gain Gold" sub-tasks on the website; however, this issue has since been rectified. The datasets pertinent to the "Gain Gold" sub-task can be acquired by downloading them from the lower section of the following website: [https://sites.google.com/view/hok-offline/home/hok-3v3](https://sites.google.com/view/hok-offline/home/hok-3v3). Similarly, the datasets pertaining to the "Destroy Turret" sub-task can be accessed and obtained from this web address: [https://sites.google.com/view/hok-offline/home/hok-1v1](https://sites.google.com/view/hok-offline/home/hok-1v1).
>
> ●**Q3: Does level-based multi-task align with the multi-task setting?**
>
> We would like to clarify this question from two distinct perspectives.
>
> Firstly, it's demonstrated from the game replay that the skills for defeating opponents of different levels vary. For instance, when facing lower-level opponents, the stronger side tends to engage in direct combat to swiftly defeat them, showcasing aggression. On the other hand, when facing opponents of similar or higher level, heroes prioritize early development to secure an economic advantage, which is significant in MOBA games, exhibiting prudence. Consequently, defeating opponents at different levels requires the model to possess multiple fight strategies simultaneously, which is meaningful in multi-task settings.
>
> Furthermore, we can conceptually validate our approach by examining  the concept of multi-task reinforcement learning. Firstly, within the context of Honor of Kings (HoK), the opponents can be considered integral components of the overall environment, and in the context of varying opponent levels, opponent level fluctuations impact the transition $P(s' | s, a)$ because of changes in the strategies of the opponent. Additionally, in Section 2.1 of [1] and Section 3 of [2], the concept of multi-task reinforcement learning is explained. A "task" is defined as $T_i = (S, A, P_i, R_i, \gamma)$, where each task is characterized by different transition probabilities $P_i$ or reward functions $R_i$. Therefore, we can classify "level-based multi-task" within the domain of multi-task reinforcement learning due to the changes in the transition probability $P(s' | s, a)$.
>
> ●**Q4: The difference between "Level-based Multi-task" and the "mixed Multi-Difficulty" dataset proposed.**
>
> Before explaining this issue, let us recall the basic settings, which is also introduced in Section 4.3 in the paper:  in HoK, the opponents can be regarded as integral components of the environment. The observations in our datasets are all based on heroes in controlled camp, so what we need to focus on is controlled camp insted of the opponent camp.
>
> Based on the setting, in the "Level-based Multi-task" setting, the dataset comprises a fixed level model(controlled camp), engaged in combat against models of varying levels(opponent camp). During the testing phase, the performance assessment of the model is conducted by subjecting it to engagements against diverse opponent models. Hence, within this paradigm, the agent must engage in multi-task learning to acquire the skills necessary for combatting diverse opponents.
>
> Secondly, in the "Mixed Multi-Difficulty" setting, the dataset contains different strategies(poor, medium expert) models(controlled camp) fighting with a fixed model(opponent camp). During the testing stage, the model's performance evaluation is restricted to engagements solely against fixed model. Consequently, under this setting, the agent must learn the optimal strategy to effectively overcome the fixed model from a mixed dataset.
>
> **Not finished yet. Please refer to the next block.**

---

> > ### Author Response · Authors · 2023-08-10
> >
> > ●**Q5: Some of the details provided in the Appendix should be provided in the main body of the paper, i.e. a summary of the datasets.**
> >
> > Apologies for the absence of certain pertinent details within the main body of the paper. To address this shortfall, we have furnished a more comprehensive account of essential aspects in the subsequent sections in Section 4.3.
> >
> > ●**Q6: The descriptions about 'Generalization' seem to be confusing.**
> >
> > We greatly appreciate your attention to the depiction of generalization in our paper. For the purpose of testing the generalization ability of offline training models when facing new scenarios, we conducted a total of six experiments, four of which did not require additional datasets. In order to improve the comprehensiveness of our explanations, we have included additional elaboration in the second paragraph of the "Generalization"  in Section 4.3 and the second paragraph of Appendix C.1 and we have included an additional Table 7 for reference.
> >
> > ●**Q7: it indicates the framework is good at sampling diverse datasets, but does not specify why this is true**
> >
> > Sorry for the lack of detail regarding why our framework is effective at sampling diverse datasets. We have already added the descriptions of this question in Section 4.1 of our paper. Here is the brief summary:
> >
> > There are several reasons why our framework excels in sampling. Firstly, diverse datasets at different levels of expertise can be sampled by leveraging Multi-Level Models as described in Sec.4.1.1. Secondly, our framework employs parallel sampling techniques, ensuring efficient sampling of large and diverse datasets. Thank you for your feedback.
> >
> > ●**Q8: Section 4.1 is lacking in detail.**
> >
> > To improve the clarity, we have moved Section 4.3 up to earlier in the paper(now merged into Section 4.1.1), right after the decription of our framework. Besides, we have changed Figure 2 and added more descriptions about it.
> >
> > ●**Q9: Figure 2 is unclear.**
> >
> > We apologize for any confusion resulting from the unclear Figure 2 in our paper. We have enhanced the visual representation of Figure 2 by repainting it. Additionally, we have improved the clarity and accuracy of the picture's caption.
> >
> > ●**Q10: Why are the observations imperfect? How imperfect?**
> >
> > The "impefect" refers to "partial observable", which means the observaion space is partial observable and this characteristic is common especially in multi-agent settings. In HoK1v1 and HoK3v3, for example, sometimes heroes in controlled camp cannot see the position of opponent heroes or soldiers, etc. unless they appear in the sight of heroes in controlled camp, thus making the observaion space partial observable.
> >
> > For clarity, we have changed the word "imperfect" into "partial observable". Thanks for your feedback.
> >
> > ●**Q11: Is level of difficulty only based on the level of the opponents? It's ambiguous from the wording. If it's not, what other factors are there?**
> >
> > In HoK1v1 and HoK3v3 the level of difficulty also depends on some other factors, such as the different heroes. This is because some heroes skill are so complex that it's hard for agent to learn how to control them.  However, in our paper, we mainly use the fixed default hero "luban" which is comparably easy to learn, so under this setting the level of difficulty mainly based on the level of the opponents.
> >
> > ●**Q12: Lines 109-111 and 153-155 or so seem to be duplicates of each other.**
> >
> > Thanks for your feedback. We have removed the redundant lines in Section 4.2.1.
> >
> > ●**Q13: 1000 trajectories seems like a relatively small amount of data for such complex tasks**
> >
> > The choice of 1000 trajectories is based on historical experience and experimental findings. Each trajectory is typically longer than 2000 steps, resulting in a total of more than 2 million steps per dataset. This exceeds the size of typical offline datasets used in previous studies. Moreover, experimental results demonstrate that mainstream baselines deliver satisfactory performance, which suffices for benchmarking purposes. Although larger datasets may yield improved performance, they also occupy more disk space and thus are not neccessary. To ensure efficiency and portability, we opt for 1000 trajectories in each dataset. Furthermore, in future studies, we intend to conduct an thorough study on the impact of dataset size and provide larger datasets for larger models or general agents in the HoK.
> >
> > **Not finished yet. Please refer to the next block.**

---

> > > ### Author Response · Authors · 2023-08-10
> > >
> > > ●**Q14: Other than daily active users, I'm curious if there were other reasons for selecting Honor of Kings over other MOBAs. Not that I am objecting to its choice, merely curious if it provides mechanics/settings that are of more interest.**
> > >
> > > There are some other strengths for the Honor of Kings.
> > >
> > > Firstly, the magnitude of states and actions involved in HoK1v1 can reach to $10^{600}$ and $10^{18000}$ [3], exceeding the total number of atoms in the entire universe ( $10 ^{80}$ ). This makes exploration extremely difficult. In addition, the observation space is partial observable, making it unable to provide complete global real-time information like Go.
> > >
> > > Secondly, compared to other MOBA games, the interface for research of Honor of Kings is quite complete(open-sourced at https://github.com/tencent-ailab/hok_env), making it easy for researchers to use. Besides, other popular MOBA games such as Dota and League of Legends are both PC Games. In comparison, Honor of Kings is a mobile game that presents a challenging environment for in-depth research, yet it has fewer game rules than Dota and League of Legends, leading to lower resource requirements for training purposes. This characteristic makes Honor of Kings suitable for further researches.
> > >
> > > Overall, Honor of Kings is a meaningful and user-friendly open-sourced environment for researchers to evaluate the performance of algorithms, without losing its practicality.
> > >
> > > ●**Q15: Maintenance plan.**
> > >
> > > The maintenance plan for our datasets and benchmarks has been added in Appendix A.
> > >
> > > **Reference**
> > >
> > > [1] Yoo M, Cho S, Woo H. Skills Regularized Task Decomposition for Multi-task Offline Reinforcement Learning[J]. Advances in Neural Information Processing Systems, 2022, 35: 37432-37444.
> > >
> > > [2] Yang R, Xu H, Wu Y, et al. Multi-task reinforcement learning with soft modularization[J]. Advances in Neural Information Processing Systems, 2020, 33: 4767-4777.
> > >
> > > [3] Ye D, Liu Z, Sun M, et al. Mastering complex control in moba games with deep reinforcement learning[C]//Proceedings of the AAAI Conference on Artificial Intelligence. 2020, 34(04): 6672-6679.

---

> > > > ### Comment · Reviewer_D3sf · 2023-08-31
> > > >
> > > > Thank you for your detailed reply, addressing my concerns thoroughly. I have raised my score from a 5 to a 7.

---

> ### Author Response · Authors · 2023-08-25
> **Continuing the discussion**
>
> Dear Reviewer D3sf:
>
> We have observed that you have not yet engaged in the discussion, and we would like to inquire if you have any further comments or questions that we can address collaboratively. Your expertise and insights would greatly assist us in improving the paper, and we are eager to address any concerns you may have.
>
> Kindly inform us of any thoughts or questions you may still have, and we anticipate the continuation of our conversation.
>
> Best regards,
>
> Authors of Submission119

---

### Decision · Program_Chairs · 2023-09-22

**Decision:**

Accept (Poster)

**Comment:**

This work presents a novel set of pre-collected datasets for use in Offline RL and Multi-Agent Offline RL, taken from demonstrations for Honour of Kings, the world's most widely played Mobile Online Battle Arena (MOBA).  Along with presenting the datasets, which are available for certain tasks, the other contributions of the work include the benchmarking of various offline RL and offline MARL algorithms.

**Perceived Strengths:**
* The reviewers generally agreed on the utility of such a dataset in advancing research into Offline RL and MultiAgent Offline RL.  The complexity of the game and the MOBA aspect of the game were highlighted as reasonable source of complex, interesting behaviours.
* The code, data and documentation of the submission were commended for being ‘comprehensive’ and appear to be high quality.

**Perceived Weaknesses:**

* One reviewer raised concerns around a lack of diversity in the benchmark, given that the environment always revolves around one game.  Authors responded arguing that, despite being only one game, HoK poses offers significant of complexity and diversity to be a challenge in multiple areas of research including generalization, multitask learning.
* Some criticisms were made of the paper around clarity, and explaining motivation, which have been addressed in comments or in the paper.

**AC View**

Following on from the reviews and discussions, I recommend acceptance of this work on account of the accepted utility of this work to researchers in offline RL and offline MARL.  This decision was made easier thanks to the reviews and substantial responses made over the course of the review period, and the changes to the paper which I think greatly improved the clarity.

While I share concerns that 1000 trajectories for each task may be too few for such complex tasks, I believe taken as a whole, the dataset is convincing in offering opportunities to research learning complex (multi-agent) behaviours from data in a widely-popular 'real world' environment.  I do not share the opinion of Reviewer KDwt, that such a single environment does not contain enough diversity for relevant RL research and agree with Reviewer D3sf that the different datasets span a variety of compelling research directions, with ample scope for expansion.